# Long-term measurements (2010 - 2014) of carbonaceous aerosol and carbon monoxide at the Zotino Tall Tower Observatory (ZOTTO) in central Siberia

Eugene Mikhailov[1,2], Svetlana Mironova[2], Gregory Mironov[2], Sergey Vlasenko[2], Alexey Panov[3], Xuguang Chi[4], David Walter[1], Samara Carbone[5], Paulo Artaxo[5], Martin Heimann [6], Jost Lavric[6], Ulrich Pöschl[1], and Meinrat O. Andreae[1,7]

[1]Multiphase Chemistry and Biogeochemistry Departments, Max Planck Institute for Chemistry, P.O. Box 3060, 55020 Mainz, Germany

[2]Department of Atmospheric Physics, Saint-Petersburg University, St. Petersburg State University, SPbSU, SPbU, 7/9 Universitetskaya nab., St. Petersburg, 199034, Russia

[3]Sukachev Institute of Forest, Siberian Branch, Russian Academy of Sciences, 50 Akademgorodok, 660036, Krasnoyarsk, Russia

[4]School of Atmospheric Sciences, Nanjing University, 210023, Jiangsu, China

[5]Instituto de Física, Universidade de São Paulo (USP), Rua do Matão, Travessa R, 187, CEP 05508-900, São Paulo, SP, Brasil

[6]Max-Planck-Institute for Biogeochemistry, Hans-Knöll-Str. 10, 07745 Jena, Germany

[7]Scripps Institution of Oceanography, University of California San Diego, La Jolla, CA 92093, USA

*Correspondence to:* Eugene F. Mikhailov (eugene.mikhailov@spbu.ru)

**Abstract.** We present long-term (5-year) measurements of particulate matter with an upper diameter limit of ~10 μm (PM10), elemental carbon (EC), organic carbon (OC), and water-soluble organic carbon (WSOC) in aerosol filter samples collected at the Zotino Tall Tower Observatory in the middle-taiga subzone (Siberia). The data are complemented by carbon monoxide (CO) measurements. Air mass back trajectory analysis and satellite image analysis were used to characterize potential source regions and the transport pathway of haze plumes. Polluted and background periods were selected using a non-parametric statistical approach and analyzed separately. In addition, near-pristine air masses were selected based on their EC concentrations being below the detection limit of our thermal/optical instrument. Over the entire sampling campaign, 75% and 48% of air masses in winter and in summer, respectively, and 42% in spring and fall are classified as polluted. The observed background concentrations of CO and EC showed a sine-like behavior with a period of 365 ± 4 days, and mostly reflect different degrees of dilution and removal of polluted air masses arriving at ZOTTO from remote sources. Our analysis of the

near-pristine conditions shows that the longest periods with clean air masses were observed in summer, with a frequency of 17%, while in wintertime only 1% can be classified as a clean. Against a background of low concentrations of CO, EC, and OC in the near-pristine summertime it was possible to identify pollution plumes that most likely came from crude oil production sites located in the oil-rich regions of Western Siberia.

Overall, our analysis indicates that most of the time the Siberian region is impacted by atmospheric pollution arising from biomass burning and anthropogenic emissions. A relatively clean atmosphere can be observed mainly in summer, when polluted species are removed by precipitation and the aerosol burden returns to near-pristine conditions.

## 1. Introduction

The Siberian forests cover about 70% of the total area of the Eurasian boreal forest and are an important factor controlling global and regional climate. In turn, climate change causes a response of Siberian ecosystems, which shows up in a redistribution of matter and energy between terrestrial ecosystems and the atmosphere (Goetz et al., 2007; Lappalainen et al., 2016). The atmospheric aerosol over Siberia is of particular interest for several reasons. Firstly, biogenic emissions of volatile organic compounds (VOC) from the vast boreal taiga forest are thought to lead to the formation of secondary organic aerosol (SOA) (Tunved et al., 2006). Secondly, Siberia has been documented to be an important source region of biomass-burning aerosol particles that are distributed around the globe in the free troposphere (Conard and Ivanova, 1997; Müller et al., 2005; Warneke et al., 2009). Thirdly, Siberia is one of the few possible background regions in the Northern Hemisphere where near-pristine conditions prevail for certain periods of the year (Chi et al., 2013). Such atmospheric observations in remote areas are very important for providing a reference for evaluating anthropogenic impacts in this and other regions (Andreae, 2007; Carslaw et al., 2013; Spracklen and Rap, 2013).

Despite the relatively remote location and low population density, human impacts on Siberian ecosystems are increasingly noticeable. This includes expansion of agriculture at the southern end of the boreal forest zone, forest logging, as well as impacts on steppe and forest fire regimes (Chi et al., 2013; Vasileva et al., 2011; Heintzenberg et al., 2013; Mikhailov et al., 2015a, Panov et al., 2015). In addition, the massive expansion of oil and gas wells, predominantly in northwestern Siberia, is likely to have an impact on aerosol and trace gas emissions through gas flaring and potential leaks in the mining and transport infrastructure (Stohl et al., 2013; Heimann et al., 2014).

Typically, two classes of carbonaceous aerosol are commonly present in ambient air - elemental carbon (EC) (often referred as black carbon or soot; Andreae and Gelencsér, 2006) and organic carbon (OC). Carbon-containing components (EC and OC) account for 10% to 70% of atmospheric PM mass (Turpin et al., 2000; Zhang et al., 2007). Both OC and EC are important agents in the climate system, which affect the optical characteristics and thermal balance of the atmosphere both directly, by absorbing and scattering incoming solar radiation, and indirectly, by modifying cloud properties (Haywood and Boucher, 2000; Andreae and Merlet; 2001; Pierce et al., 2007, Bond et al., 2013; Andreae and Ramanathan, 2013; Alonso-Blanco et al., 2014). EC is the major light-absorbing component of atmospheric aerosol and is the second most important global warming agent after carbon dioxide (Bond et al., 2013). EC aerosols are primary submicron particulate matter emitted during incomplete combustion of fossil or bio-fuels. Aged EC aerosols have fractal-like or globular structure and a wide variety of organic and inorganic species adsorbed to them. OC is mainly a light scattering component, however is also known to have a light-absorbing fraction called "brown carbon" (BrC) that, similar to BC, has been shown to be an important factor in aerosol radiative forcing (Andreae and Gelencsér, 2006; Andreae and Ramanathan, 2013; Saleh et al., 2014). In addition, due to the lensing effect, organic coatings can substantially modify the optical properties of EC in mixed particles. Thus, laboratory experiments have shown that the light absorption enhancement factor for internal mixtures of EC with organic and inorganic coatings can range from 1.3 to 3.5 (Schnaiter et al., 2005; Mikhailov et al., 2006; Zhang et al., 2008; Shiraiwa et al., 2010; Pokhrel et al., 2017).

Forest fires and biogenic emissions from coniferous trees and forest litter are the main sources of carbonaceous aerosols emitted into the atmosphere over boreal forests. The molecular composition of OC from biomass burning is still poor understood. It is known only that the thermal decomposition products of plant cellulose and lignin provide the largest contribution to the organic mass. Among them are sugar anhydrides, phenol compounds, alcohols, esters, aldehydes and ketones (Kanakidou et al., 2005; Hoffer et al., 2006; Lappalainen et al., 2009; Mayol-Bracero et al., 2002; Fuzzi et al., 2007). In the spring and summer seasons, biogenic activity increases and volatile organic compounds (VOCs), mainly isoprene, terpenes, and sesquiterpenes, are emitted to the atmosphere. These compounds form secondary organic aerosols as a result of photochemical oxidation and gas-to-particle conversion (Hallquist et al., 2009). A substantial fraction of boreal forest organic aerosol consists of water-soluble compounds and therefore contributes to hygroscopic growth and cloud condensation nuclei (CCN) activity (Huff Hartz et al., 2005; Ehn et al., 2007; Carrico et al., 2008; Cerully et al., 2011; Jaatinen et al., 2014; Paramonov et al., 2013; Mikhailov et al., 2015b).

Primary biological aerosols (PBA) and their rupture products are another subset of organic particles, which are directly released from the biosphere into the atmosphere in the growing season. They comprise living and dead organisms (e.g., algae, archaea, bacteria), dispersal units (e.g., fungal spores and plant pollen), and various fragments or excretions (e.g., plant debris, brochosomes, and salt particles) (Despres et al., 2012; Fröhlich-Nowoisky et al., 2016; Pöhlker et al., 2012). Their particle sizes range from nanometers up to about a tenth of a millimeter. They can serve as nuclei for cloud droplets, ice crystals, and precipitation, thus influencing the hydrological cycle and climate. In pristine air over vegetated regions, bioaerosols are likely to be an essential regulating factor in the formation of precipitation (Pöschl et al., 2010; Pummer et al., 2012, 2015; Steiner et al., 2015). In the atmosphere, bioparticles undergo internal and external mixing with other aerosols, including SOA, which can influence bioaerosol properties through SOA coatings (Hallquist et al., 2009; Pöschl et al., 2010; Huffman et al., 2012; Pöhlker et al., 2012).

In the northern latitudes of Eurasia, climate change occurs 1.5-2 times faster than the global average (Hansen et al., 2006; Groisman and Soja, 2009; IPCC, 2014). One consequence of these changes is an increase of VOC emissions, which is a precursor of SOA formation (Tunved et al., 2006). It is expected that the increase in the concentration of biogenic aerosols will provide negative climate feedbacks, involving aerosol-cloud and aerosol-carbon cycle interactions (Kulmala et al., 2004, 2014; Andreae et al., 2008a). These feedbacks may change the local structure of the atmospheric circulation, cloud properties, and the intensity of precipitation (Rosenfeld et al., 2013; Lappalainen et al., 2016). Precipitation, in turn, causes emissions bursts into the atmosphere of primary bioaerosols and submicron degradation products containing hygroscopic water-soluble inorganic ions (potassium, sodium, chlorides, and phosphates) and polysaccharides (hexoses, mannitol) (Morris et al., 2014; Bigg et al., 2015; Huffman et al., 2013). As temperatures increase, population outbreaks of tree-damaging insects can occur more frequently. The emissions of organic substances from damaged trees are significantly higher than from healthy trees (Bergström et al., 2014). For example, in the boreal environment, trees damaged by the pine borer (*Neodiprion sertifer*) emit 11 times more monoterpenes and 20 times more sesquiterpenes compared to healthy trees. As a result, the total mass of aerosols increased by 480% (local maximum), and the concentration of cloud condensation nuclei increased by 45% (Joutsensaari et al, 2015). It is also projected that increased growth of the forest area resulting from the surface temperature increase will be accompanied by more frequent forest fires (Shvidenko et al., 2011), which are powerful sources of aerosol particles and greenhouse gases (Paris et al., 2009; Janhäll et al., 2010; Smolyakov et al., 2014). In addition, industrial production and steady increase in oil

and gas production in Siberia will further increase the concentrations of aerosols in the Siberian air basin. It is therefore expected that the role of biogenic, pyrogenic, and anthropogenic aerosol emissions will grow with increasing temperatures and their influence on climate change will be important both on the regional and global level (Kulmala et al., 2014; Lappalainen et al., 2016). In order to assess the magnitude and sign of these climatic effects, as well as to predict the possible consequences for Siberian ecosystems, it is necessary to provide a comprehensive long-term monitoring of the burden and composition of atmospheric aerosols in the region.

In 2006, the 300-m tower of the Zotino Tall Tower Observatory (ZOTTO) was established in Central Siberia (Heimann et al., 2014). The background character and the geographical location of this station are appropriate conditions for studying atmospheric transport and coincident chemical transformation of polluted air at a wide range of spatial and temporal scales, particularly for assessing the potential influence of emissions from various natural and anthropogenic sources on surface air composition over the large territory of Siberia. The remote location of ZOTTO in the middle of the Siberian taiga forest also makes it highly suitable for investigating the exchange of trace gases with this ecosystem and the production of aerosol by the boreal forest. Continuous measurements of comprehensive sets of atmospheric constituents in the gas and particle phase together with meteorological parameters have been carried out at ZOTTO since October 2006 (Kozlova et al., 2008; Heintzenberg et al., 2008). With regards to aerosol, a study of the representativeness of the ZOTTO facility and first analyses of the particle size distribution data can be found in Heintzenberg et al. (2008) and Heintzenberg and Birmili (2010); a statistical analysis of particle size distribution, particle absorption, and carbon monoxide (CO) data taken from the first four years of operation of the ZOTTO facility (September 2006 to January 2010) is given in Heintzenberg et al. (2011), together with seasonally dependent major air mass pathways and the related particle size distributions. An extended statistical analysis of aerosol properties including scattering coefficients, Ångström exponents, single scattering albedo, and backscattering ratios, as well as a CO data set with seasonal, weekly, and diurnal variations between September 2006 and December 2011can be found in Chi et al. (2013).

In 2010, a filter-based sampler was mounted at the ZOTTO station for aerosol chemical analysis. In this study, we present the time series of carbonaceous aerosol measurements coupled with CO data for 5 years (2010 – 2014). We investigate the seasonal variations of EC, OC, WSOC (water soluble organic carbon), and CO. We analyze polluted, background, and near-pristine periods as well as the most pronounced pollution events and their sources observed over the entire sampling campaign. The methodology is described in Sect. 2, and the seasonal features of temporal variations and air mass origins during pollution periods are discussed in Sect. 3.4

and 3.5. The background and near-pristine air masses and their characteristics are discussed in Sect. 3.6.

## 2. Methods

### 2.1 Aerosol sampling

The aerosol samples were collected from April 2010 to June 2014 at the Zotino Tall Tower Observatory facility, which is located near the Yenisei river at the eastern edge of the West Siberia Lowland in the boreal zone (60.8 °N and 89.4 °E, 114 m asl), about 600 km north of the closest large city, Krasnoyarsk (950,000 inhabitants); the nearest village (Zotino) is about 20 km east of the site. The site lies in a vast region of boreal coniferous forest and bogs, and the ecosystem in the light taiga around the station is dominated by *Pinus sylvestris* forest stands (about 20 m height) on lichen covered sandy soils. The heart of the station is a 300-m tower, which was designed for long-term atmospheric observations and where the sampled air masses are representative of a very large fetch area. A more detailed description of the ZOTTO facility is given elsewhere (Heimann et al., 2014). The climate is dominated by a large seasonal temperature cycle reaching from minima below −55 °C in winter to maxima above 30 °C in summer.

Ambient air was sampled through a stainless steel inlet pipe with an internal diameter of about 2.9 cm, reaching to the top of the tower at 300 m above ground. The inlet was designed for a laminar nominal sampling flow of 40 L min$^{-1}$ (Birmili et al., 2007). Pre-installation calibration showed that particles with diameter $D_p > 50$ nm are nearly perfectly transmitted through this pipe (Heintzenberg et al., 2008). Additional test measurements with supermicron aerosol particles have shown that the upper transmission size limit for the inlet system is ~10 μm. Thus, carbonaceous species concentrations obtained in this study refer to aerosol particles with an upper limit of ~10 μm (PM10).

Aerosols were collected directly from the inlet line on 47-mm quartz fiber filters (2500QATUP, Pallflex) at a flow rate of 20 L min$^{-1}$ using a home-made sampler. One of the difficulties in long-term filter sampling is the decision about the length of sampling periods. On one hand, the aerosol concentration at ZOTTO is very low during the near-pristine periods, therefore days or even weeks of sampling time are needed in order to obtain enough material on the filter for analysis; on the other hand, extremely high aerosol concentrations were observed during pollution episodes (e.g., biomass burning) and higher time resolution was needed for better characterization of such episodes and to avoid overloading of the filters. As a result, the sampling time for each filter varied from 10 hours during pollution events to 480 hours during clean periods, and the median sampling time was 104 hours. A total of 292 samples were collected between

April 2010 and June 2014. The exposed filters were sealed in aluminum foil and then placed in Ziploc bags. The samples were stored at -18 ℃ before being analyzed.

### 2.3 Instrumentation

### 2.3.1 Carbon monoxide measurements

CO was measured by UV resonance fluorescence, using a Fast-CO-Monitor (model AL 5002, Aerolaser GmbH, Germany). Details of the experimental setup and calibration are described elsewhere (Chi et al., 2013). The original CO data, measured with a frequency of 3 s, were converted to 1-h averages to minimize uncertainties inherent in the data analysis methodology. These CO concentrations were further averaged over the aerosol sampling intervals. Technical problems occurred with the CO monitor during the following periods, resulting in gaps in the CO time series (Fig. 2): 7 July to 20 September 2010, 30 December 2010 to 11 April 2011, 28 June to 14 July 2011, 14 November to 27 December 2011, 17 February to 9 March 2012, and 15 May to 14 June 2012.

### 2.3.2 Gravimetric PM measurement

The aerosol mass concentrations were determined gravimetrically using a Mettler-Toledo micro balance model XP6 with 0.6 µg sensitivity. Before being weighed, the filters were equilibrated for 24 h at a constant temperature of 23 °C and a relative humidity between 35 and 45 %. Each filter was weighed at least three times before and after sampling. An anti-static U-Electrode (Mettler-Toledo) was used to remove electrostatic charge before weighing. The uncertainty (1 standard deviation) for the PM determination is estimated to be 10 µg for 47-mm quartz filters.

### 2.3.3 Organic carbon and elemental carbon analysis

OC, EC, and total carbon (TC = OC + EC) were measured by a thermal-optical transmission (TOT) technique (Birch and Cary, 1996), using a thermal-optical carbon analyzer from Sunset Laboratory (OR, U.S.A.). The temperature protocol used was NIOSH5040 (National Institute for Occupational Safety and Health) with a preset maximum of 870 ℃ (Birch, 1998). The uncertainty in the OC, EC, and TC measurements is provided for each individual filter sample by the calculation program. The uncertainty is made up of a constant part (which is 0.2 µg C cm$^{-2}$ for OC and EC and 0.3 µg C cm$^{-2}$ for TC) and of a variable part, which amounts to 5% of the OC, EC, or TC mass loading. To correct for the positive artifact in the OC determination, two quartz filters in series were used (Maenhaut and Claeys, 2007). Both filters were pre-baked at 850 ℃. The

carbon loading on the second filter was subtracted from that on the first filter. WSOC was determined by soaking part of the filter in water (18.2 MΩ cm, Direct-Q3 UV, Millipore) for 12 hours; after drying the remaining carbon in the filter was measured using the Sunset instrument. In addition to the TOT method, a TOC-V$_{CPH}$ analyzer (5000 A, Shimadzu) was also used for

WSOC analysis. A two-step procedure consisting of measurements of water-soluble total carbon (WSTC) and water soluble inorganic carbon (WSIC) was applied. WSOC is then calculated as a difference between WSTC and WSIC (Chi et al., 2009). The TOT and TOC-V$_{CPH}$ measurements of WSOC concentrations cover the date range from April 2010 to December 2011. In general, the agreement between the two methods during this time period was within 10%. Due to fatal

technical problems with the TOC-V$_{CPH}$, after December 2011 WSOC was measured only by the Sunset instrument. The estimated error of the WSOC concentrations using the TOT method is 10% - 15%, depending on the filter loading, which results in a 12-17% error for the WSOC/OC ratio.

Organic matter (OM) was estimated as 1.8·OC. The same OC-to-OM conversion factor of

1.8 had been used in the SMEARII (Finland) (Maenhaut et al., 2011a) and K-puszta (Hungary) (Maenhaut et al., 2008) remote coniferous forest sites, providing the best agreement in the aerosol chemical mass closure calculations. As a result, the total carbonaceous matter (TCM) was calculated as TCM = 1.8·OC+EC. It should be noted that there is considerable variability in reported OM/OC ratios for organic compounds depending on the relative contribution of primary

and secondary organic aerosol sources, with reported values ranging from 1.2–2.4 (Turpin and Lim, 2001). In this study OM and TCM are estimated and used mainly to illustrate their temporal variability. However, as will be shown below, the obtained estimates of the TCM/PM10 ratio are reasonably consistent with published values for the sources of the pollution plumes.

It needs to be noted that the OC/EC analysis is very sensitive to the temperature protocol

used and to the optical correction method (OC/EC split). NIOSH and IMPROVE (Interagency Monitoring of Protected Visual Environment) are the most widely applied thermal protocols, which differ in their temperature ramping regime and charring correction. The discrepancy between NIOSH- and IMPROVE-derived EC concentrations may vary in the range of a factor of 1.2 - 2, depending on the source and aging of the samples (Chow et al., 2001; Cheng et al., 2014;

Wu et al., 2016). Therefore, wherever possible, we compare OC/EC results from studies using the same or similar analytical methods.

In addition to the thermal-optical method, a single-wavelength (574 nm) Particle Soot Absorption Photometer (PSAP, Radiance Research, Seattle, USA) was used in this study for measuring the particulate light absorption coefficient to estimate equivalent black carbon (BC$_e$) con-

centrations (Andreae and Gelencsér, 2006). Details about this method and the methodology used for data analysis and interpretation correction can be found in Chi et al. (2013). In this study, PSAP and TOT measurements cover the date range from 19 April 2010 to 15 May 2012, therefore the PSAP data will be mainly used to quantify the correction factor ($BC_e$/EC ratio) between the two methods used. The $BC_e$ mass concentration ($\mu g\ m^{-3}$) can be calculated based on the relation:

$$\sigma_{abs} = \alpha_{abs}BC_e ,\tag{1}$$

where $\sigma$ ($Mm^{-1}$) is the PSAP-measured absorption coefficient, and $\alpha_{abs}$ ($m^2\ g^{-1}$) is the mass absorption efficiency. The commonly used value of $\alpha_{abs}$ is $10\ m^2\ g^{-1}$, as recommended by the PSAP manufacturer in their manual.

## 2.4 Ancillary products

Air mass trajectories were calculated with the Hybrid Single-Particle Lagrangian Integrated Trajectories (HYSPLIT) model (Stein et al., 2015) using the NCEP/ NCAR meteorological archive data produced by the National Center for Environmental Prediction (NCEP) and the National Center for Atmospheric Research (NCAR) ($2.5^{\circ}$ horizontal resolution, 17 pressure levels) (Kalnay et al., 1996). The trajectories were calculated for an arrival height of 300 m above ground level, which corresponds to the aerosol sampling height at the ZOTTO station. The NCEP/NCAR meteorological dataset was also used to estimate the mixing layer depth (MLD) and accumulated precipitation along the trajectory (APT). We used the APT data as a measure of wet removal of aged carbonaceous species (Kondo et al., 2011a; Matsui et al., 2011; Kanaya et al., 2016). In some cases, the APT values were averaged by dividing the total precipitation along all the trajectories for a selected time period by the number of these trajectories (Table 4).

Fire (hot spot) images were obtained from the Moderate Resolution Imaging Spectroradiometer (MODIS) instruments at 1 km resolution (Giglio et al, 2003) distributed by the Fire Information for Resource Management System (FIRMS). The probability of fire detection is strongly dependent upon temperature and area. Thus, local fires with a combustion temperature above 1000°C can be detected with flaming areas less than $100\ m^2$ while smoldering fires are difficult to detect (Giglio et al, 2003). Particularly, MODIS products have been successfully used for monitoring of the flare sites associated with production of crude oil (Elvidge et al., 2011; Anejionu et al., 2015).

## 3. Results and discussion
### 3.1 PM, TCM, and regional meteorology

The time series of PM10 and TCM concentrations together with daily averaged meteorological parameters during the sampling period are shown in Fig. 1. Visual analysis indicates that the highest concentrations of PM10 and TCM, observed in the spring and summer periods (Fig. 1a), correspond to high temperatures and lower levels of precipitation (Fig. 1b). It is also seen that TCM was the dominant species in particulate mass during summertime. As will be shown below, biomass burning is the main source of polluted air masses arriving at the ZOTTO site in the summer season. Our overall median PM10 concentration (25[th]-75[th] percentiles) was 7.9 (5.1-14.4) $\mu$g m$^{-3}$. This is comparable to the annual-mean PM10 values ranging from 8 to 16 $\mu$g m$^{-3}$ at Northern European rural background stations reported by Querol et al. (2004), but slightly higher than the annual-mean European continental background PM10 concentration of 7.0±4.1 $\mu$g m$^{-3}$ obtained by Van Dingenen et al. (2004).

Our measurements of the carbonaceous species can be compared with those from the Hyytiälä (Finland) and K-puszta (Hungary) remote European forested sites, which used the same EC/OC thermal protocol (Chi, 2009; Maenhaut et al., 2011b). Table 1 shows that the summertime EC concentration at ZOTTO (0.13 $\mu$g m$^{-3}$) is comparable to measurements at Hyytiälä (0.12 $\mu$g m$^{-3}$) and K-puszta (0.16 - 0.19 $\mu$g m$^{-3}$), whereas the median OC and WSOC concentrations at ZOTTO are higher than those at the European sites. Elevated concentrations of the organic species at the ZOTTO site can be attributed to the strong influence of biomass burning events during the summer season. This is particularly reflected in the wide concentration range of OC and WSOC, varying from 0.3 to 106 $\mu$g m$^{-3}$ and 0.2 to 45 $\mu$g m$^{-3}$, respectively (Table 1). In summertime, the WSOC/OC ratio at the ZOTTO site was 0.65 ± 0.13. This value is close to that for the Hyytiälä boreal forest site (0.62 ± 0.09) and slightly higher than for the K-puszta station (0.57 ± 0.11) (Table 1). The obtained WSOC/OC ratio is typical for nonurban sites surrounded by pine forest. It has been documented that in the summer season aged secondary organic aerosols produced by monoterpene oxidation as well as biomass burning organic particulate matter contain a high fraction of WSOC, ranging from 0.5 to 0.8 (Saxena et al., 1995; Kiss et al., 2002; Pöschl, 2005; Pio et al., 2007; Timonen et al., 2008).

Table 1 also provides summary statistics for carbonaceous species at ZOTTO compared with those reported for high-altitude remote sites. The annual average value of EC is within the concentration range observed at the European high-altitude sites and lower than those from Chinese stations, whereas the yearly-average OC and WSOC concentrations at ZOTTO noticeably exceed those obtained at both the European and Chinese sites. As a result, our EC/TC ratio is lower than those at the high-altitude stations, probably because of higher contributions from

VOC oxidation at ZOTTO. Note, as mentioned above, that the EC discrepancy can be partially caused by the different thermal protocols used for EC/OC analysis.

Overall, the comparative analysis of the carbonaceous species concentrations suggests that the aerosols sampled at ZOTTO during our study period were generally representative of a fairly clean region.

### 3.3 Estimation of polluted, background, and clean periods

Siberia is a highly diverse region, where relatively clean periods (PM10 < 1 μg m$^{-3}$) alternate with heavily polluted intervals (PM10 > 50 μg m$^{-3}$) (Fig.1a). Air masses that are influenced by regional emission sources are not representative for well-mixed background air. Data filtering is therefore an important step in analyzing the data. In our analysis, we therefore differentiate polluted, background, and clean periods.

We refer to "background" conditions as an atmospheric state without the detectable influence of local or regional pollution sources, but affected by emissions from natural origin as well as by pollution transported from very distant sources (Andreae, 2007; Chi et al., 2013; Mikhailov et al., 2015b). To define the concentrations of the measured species representing background air (or the other way around to define the polluted periods) we made use of a non-parametric statistical approach named REBS (Robust Extraction of Background Signal). This technique has been previously applied for the identification of background CO concentrations for long-term measurements at the high alpine background site at Jungfraujoch (Ruckstuhl et al., 2012), at a global background station in China (Zhang et al., 2011), and at the ZOTTO site (Chi et al., 2013). Here, we use a bandwidth of three months following the suggestion of Ruckstuhl et al. (2012). Figure 2 shows the time series of the CO mixing ratio, EC, OC, and PM10 concentrations at ZOTTO, with the red baseline indicating REBS background concentrations $g_i(t_i)$. All concentrations $C_i(t_i)$ ≤ $g_i(t_i)+3\sigma$ are classified as "background" measurements; all other observations are classified as "polluted". Here, $\sigma$ is the goodness of fit (Ruckstuhl et al., 2012).

In contrast to background conditions, the term "pristine" implies that all aerosol sources arise from natural emissions. It has been argued that regions in which aerosols are totally unperturbed by air pollution no longer exist in today's atmosphere (Andreae, 2007). Therefore, the terms "near-pristine" or "clean" are commonly used to denote relatively clean air. Both EC (or BC) and CO are indicators of combustion and pollution, and their concentrations are frequently used to separate near-pristine from polluted periods (Andreae, 2007; Chi et al., 2013; Hamilton et al., 2014). It should be noted, however, that it is not always possible to define a pristine environment based on CO concentrations at a particular site, since CO can be produced by biogenic

sources directly or by means of $CH_4$ and NMVOCs (nonmethane volatile organic compounds) oxidation (Gaubert et al., 2016 and references therein). The concentration of EC (or BC) is more reliable for air quality analysis, as EC is a primary pollutant, emitted directly into the atmosphere during incomplete combustion of carbonaceous fuels. During long-range transport it can be removed by precipitation scavenging together with other pollution species, returning the aerosol burden to near-pristine conditions. Therefore, in this study the periods with EC concentrations below the limit of detection ($0.02\pm0.01$ $\mu gC\ m^{-3}$) are classified as clean or near-pristine.

Table 2 shows the seasonal concentrations of CO, PM10, and carbonaceous species for polluted, background, and clean conditions averaged over the full sampling period (2010-2014). We compared our carbon monoxide concentrations with those obtained at the ZOTTO site from 2006 to 2011 by Chi et al. (2013). During background and clean periods the CO concentrations are in agreement within about 10%, while the polluted CO mixing ratios disagree markedly in all seasons, especially in the summer and fall periods, where the averaged concentrations exceed those obtained by Chi et al. (2013) by factors of 6.5 and 1.6, respectively. As a consequence, the frequencies of polluted periods are higher than those obtained in the Chi et al. (2013) study. Accordingly, the frequencies of clean periods are significantly lower (Table 2). Besides more strong pollution events observed from 2010 to 2014, our low frequencies of clean periods also arise from a more rigid criterion used for clean period detection. In Chi et al.'s (2013) work, clean periods were selected based on the aerosol absorption coefficients, i.e., the time intervals when their values were below 1 $Mm^{-1}$ (~ 0.1 $\mu gC\ m^{-3}$) were classified as clean. In contrast, in this study the clean periods were defined by EC concentrations below the detection limit of $0.02\pm0.01$ $\mu gC\ m^{-3}$.

Table 2 also shows that in the cold period (winter, spring, and fall) the PM10, OC, and WSOC concentrations as well as the WSOC/OC and TCM/PM ratios obtained for background and clean periods show only small seasonal differences. In these seasons, aerosol particles come to ZOTTO mainly from pollution sources, since in the cold period the air temperature is below or close to 0°C and biogenic activity is suppressed. Previous studies (Chi et al., 2013) have shown that in winter, pristine-period particles and pollution particles have comparable footprint and size distribution. These authors suggested that in the cold seasons the main sources of the observed particles at the ZOTTO site are the same during clean and polluted periods, and that the different concentrations are mostly due to different degrees of dilution and removal. We concur with their suggestion that the differences between polluted and "clean" concentrations in the cold seasons (Table 2) are mainly the result of changes in meteorological conditions (wind direction, precipitations). Note also that across all seasons, the WSOC/OC ratio ranged from 0.5 to 0.7 (Table 2).

This interval is typical for aged primary and secondary organic aerosols, which have undergone chemical transformations (oxidation, nitration, hydrolysis, and photolysis) during long-range transport (Pöschl, 2005; Timonen et al., 2008; Jimenez et al., 2009; Saarnio et al., 2010).

**3.4 Characterization of polluted air**

As described in Sect. 3.3, the entire ZOTTO data set was separated into polluted and background periods using the REBS method. In order to evaluate the degree of pollution, we used the Enhancement Factor (EnF), defined here as the ratio of the median concentration of a species during a polluted period to the background concentration. The monthly variations of EnF for PM, EC, and OC are shown in Fig. 3. It is seen that the EnF of all species correlate quite well, showing high values of the EnF during the summer months and lower values in the cold season. Particularly, the EnF for EC indicates that minimal levels of pollution were observed in March and April with EnF of ~1.7, followed by the winter months (EnF~ 3), while maximal pollution is present in summer with EnF = 20, 51, and 5.4 for June, July, and August, respectively. In general, Figure 3 clearly illustrates that during the whole year the air masses arriving at ZOTTO contain pollution components from regional and local sources.

Biomass burning and fossil fuel combustion processes emit CO, OC, and EC (BC), but the emission ratio between these species differs by combustion type and burning condition (Kondo et al., 2006; Wang et al., 2011; Zhang et al., 2013). We used the $\Delta OC/\Delta EC$ and $\Delta EC/\Delta CO$ relationships observed at ZOTTO as an additional indicator of the origin of pollution emissions (Andreae and Merlet, 2001; Kondo et al., 2011a; Wang et al., 2011). Figure 4 and Figure 5 show the seasonal relationships between $\Delta EC$ and $\Delta OC$ and between $\Delta EC$ and $\Delta CO$, respectively. The pollution concentration enhancements ($\Delta$ values) were calculated as the difference between the measured concentrations and the REBS-defined background values.

Figure 4 shows strong correlations between $\Delta OC$ and $\Delta EC$ concentrations throughout all seasons, but the $\Delta OC/\Delta EC$ slopes are different. The good correlation between $\Delta OC$ and $\Delta EC$ implies that both components are emitted into to the atmosphere from the same combustion sources, while the significant seasonal differences between the regression coefficients suggest that the sources of carbonaceous aerosol are different between seasons. Depending on fire intensity and type of fire (e.g., flaming vs. smoldering) the OC/EC ratio for boreal forest fires varies over a wide range (Table 3). From prescribed burns conducted in boreal forests (northern Ontario, Canada), Mazurek et al. (1991) reported a range of 10-18 and 21-95 for full-flaming and smoldering fire conditions, respectively. A chamber experiment conducted with typical Siberian biomass (pine) showed that the OC/EC ratios for PM10 in fresh (aged) smoke for flaming and

smoldering are 0.3 (2.3) and 181(126), respectively (Popovicheva et al., 2015). In addition, the OC/EC ratios for fresh forest debris smoke were found to be 0.6 and 35 in flaming and smoldering fires, respectively. In mixed boreal wildfires, the mean OC/EC ratios for PM10 in fresh smoke plumes were reported as 6.7 (Saarnio et al., 2010), 6.5 (Popovicheva et al., 2015), and 15 (Lee at al., 2005). For aged smoke particles, Yan et al. (2008) reported a value of 25.6, which is close to our summer averaged $\Delta OC/\Delta EC$ ratio of 27.6 ± 1.0 obtained for the entire sampling campaign (Fig.4c).

In contrast to open vegetation fires, industrial biomass and fossil fuel combustion as well as domestic fuel combustion generally exhibit smaller OC/EC ratios. Close to sources, this ratio ranged from 0.7 to 5 depending on the fuel burned and the combustion technology used (Andreae and Merlet, 2001; Lim and Turpin, 2002; Andreae et al., 2008b; Saarikoski et al., 2008). It should be noted that aging of air masses tends to increase the OC/EC ratio of the aerosol due to oxidation and condensation of organic material (Burtscher et al., 1993; Andreae et al., 2008b; Konovalov et al., 2015, 2017). On the whole, our seasonal $\Delta OC/\Delta EC$ ratios are in agreement with the literature (Table 3) indicating that the low value of 6.4 ± 0.4 obtained in wintertime (Fig. 4a) is mostly the result of local and regional anthropogenic emissions from fossil fuel and domestic combustion, while the high value of 27.6 ± 1.0 (Fig. 4c) observed in summer is indicative of open biomass burning, probably with a large fraction of smoldering fires. The intermediate ratios of 12.9 and 18.2 obtained in spring (Fig. 4b) and fall (Fig. 4d) suggest that the air masses came to ZOTTO both from fossil-fuel and biomass-burning sources.

Unlike the $\Delta EC$-$\Delta OC$ relationship, the correlation between $\Delta EC$ and $\Delta CO$ is not as strong and varies greatly from season to season. The strongest correlation is found in the summer season ($R^2$=0.85; Fig. 5c), followed by the wintertime ($R^2$=0.32; Fig. 5a), while no significant relationship between $\Delta EC$ and $\Delta CO$ was observed in spring and fall, with $R^2$=0.02 (Fig. 5b) and $R^2$=0.02 (Fig. 5d), respectively. The strength of the correlation between EC and CO strongly depends on the meteorological conditions. Wet deposition can affect aged EC (Matsui et al., 2011; Verma et al., 2011; Kanaya et al., 2016) but not CO, while CO can be removed from the atmosphere by oxidation with OH radicals (Hamilton et al., 2014). In addition, as noted above, CO can be produced from the oxidation of methane and nonmethane hydrocarbons, which are mainly of biogenic origin and to lesser extent from anthropogenic sources (Gaubert et al., 2016). It seems therefore reasonable that the strength of the correlation between EC and CO depends on the season and the history of the air mass before arriving at ZOTTO. The strong correlation obtained in the summer season and the $\Delta EC/\Delta CO$ slope of 1.3±0.1 ng m$^{-3}$ ppb$^{-1}$ suggest that both components were emitted into to the atmosphere from the same nearby biomass burning sources.

EC/CO emission ratios of 1.7±0.8 and 3.4±1.6 ng m$^{-3}$ ppb$^{-1}$ for smoldering and flaming combustion phases, respectively, were obtained during summer aircraft measurements in fresh boreal forest fire plumes (Kondo et al., 2011a) (Table 3). Therefore, our low overall summer ΔEC/ΔOC ratio of 1.3 ng m$^{-3}$ ppb$^{-1}$ suggests that the air masses arriving at ZOTTO were mainly influenced by smoldering fires. The fact that smoldering emissions tend to stay within the boundary layer because of the lower buoyancy produced by the less efficient smoldering combustion, whereas flaming emissions tend to rise several thousand meters, may also have contributed to the prevalence of smoldering emissions at ZOTTO.

The ΔEC/ΔCO slope averaged over all winter polluted periods is 3.3 ± 0.8 ng m$^{-3}$ ppb$^{-1}$ (Fig.5a). This is within the range of 1.4 – 5.4 ng m$^{-3}$ ppb$^{-1}$ obtained in winter at the Hedo site (Okinawa Island, Japan) due to long-range transport of pollution plumes from East Asia (Verma et al., 2011), but on average it is lower than the values reported for industrial and urban regions of East Asia and North America (Table 3). Most likely in wintertime scavenging by ice crystal precipitation (Cozic et al., 2007) and dry deposition of EC has resulted in decreasing ΔEC/ΔCO ratios during the air mass transport from the pollution sources to the ZOTTO site.

The lack of correlation ($R^2$~ 0.02-0.03) between ΔEC and ΔCO concentrations in spring (Fig. 5b) and in fall (Fig. 5d) is an indicator of the long-range transport of the polluted air masses from diverse sources. Since in the transition seasons the secondary CO sources and sinks are small, it is reasonable to assume that wet deposition of EC as well as variable source types are responsible for the poor relationship between EC and CO. In addition, CO from long-range transport from Europe may contribute a substantial fraction of CO at the ZOTTO site (Chi et al., 2013).

## 3.5 Pollution episodes

Next we consider in detail some specific pollution events that occurred during the sampling period from April 2010 to June 2014. We selected the most prominent events, where PM10, EC, CO, and OC concentrations exceeded their REBS background level simultaneously. These selected pollution periods and summary statistics for carbonaceous species, including their ratios, are shown in Table 4. For all $\Delta X_i / \Delta X_j$ ratios presented in Table 4, each species concentration enhancement, $\Delta X_i$, was obtained as the difference between measured and REBS-derived concentrations. In addition, we also included the MLD, height above ground level (AGL), and APT averaged along the air mass trajectories.

### 3.5.1 Polluted winter air

Figure 6 shows winter polluted 5-day air mass trajectories arriving at ZOTTO in the periods of 25-30 December 2010 (Fig. 5a, orange), 10-15 January 2011 (Fig. 5a, grey) and from 27 December 2011 to 9 January 2012 (Fig. 5b). It is seen that all trajectories passed over the south-western and southern regions with densely populated industrialized areas, and therefore air masses moving across this area are likely to have accumulated anthropogenic emissions. During these periods, the trajectory heights (Fig. 5 c,d) were less than or comparable to the mixed layer depth (Table 4), indicating that the polluted air masses were trapped under a very low inversion layer. In addition, the precipitation rates along the trajectories were moderate (Table 4). These meteorological conditions contributed to the relatively high CO, PM10, and carbonaceous species concentrations observed during the winter pollution periods. As a result, the overall $\Delta OC/\Delta EC$ slope for the winter pollution periods is $3.9 \pm 0.6$ ($R^2 = 0.75$) (Table 3). This value is consistent with emission ratios found in urban areas, and represents a mixture between aged regional haze aerosols with OC/EC ranging from 3 to 7 and local emissions with a ratio of about 1.4 (Andreae et al., 2008b; Meng et al., 2007; Li et al., 2012; Cheng et al., 2011; Pan et al., 2012) (Table 3).

The $\Delta EC/\Delta CO$ slope for the winter pollution episodes is $5.8 \pm 0.7$ ng m$^{-3}$ ppb$^{-1}$ ($R^2 = 0.85$) (Table 3). This ratio is consistent with a fossil-fuel dominated carbonaceous aerosol, as is typically observed at urban sites, e.g., $5.7 \pm 0.9$ (Tokyo; Kondo et al., 2006), $5.4 \pm 0.4$ (Guangzhou; Verma et al., 2010), $5.8 \pm 1.0$ (Houston, Dallas; Spackman et al., 2008), $3.5 - 5.8$ (Beijing; Han et al., 2009), and somewhat lower than values found for periods when urban air masses are polluted by diesel traffic emissions: $7.9 \pm 0.2$ (Guangzhou; Andreae et al., 2008b), $7.7 \pm 0.2$ (Beijing, fall; Wang et al., 2011), and $8.4 \pm 0.4$ (North China Plain; Wang et al., 2011) (Table 3).

In winter, the $\Delta TCM$ fraction in the $\Delta PM$ varies from 0.17 to 0.37 (Table 2), which implies that the dominant PM10 components are inorganic compounds. Our independent measurements performed from 15 to 25 December 2011 showed that the main inorganic anthropogenic ions are sulfates (~32%), nitrates (~6%), ammonium (~5%), and minor components such as sodium (~1%) and potassium (~1%). The contribution of other ions is less than 1% (Mikhailov et al., 2015b).

**3.5.2 Polluted summer air**

Previous studies based on CO measurements during 2007 and 2008 (Vasileva et al., 2011) and those complemented by an analysis of aerosol properties (concentration, absorption and scattering coefficients) from 2006 to 2012 (Chi et al., 2013) indicate that in the summer season pollution events are mainly associated with biomass burning emissions. Our results are in agreement

with these earlier studies. Table 4 shows that within the selected summer periods, CO, PM10, and carbonaceous species concentrations are extremely high, exceeding their background values by a factor of 20 or even more (Fig. 3), and suggesting therefore that the biomass burning events were close to the ZOTTO site. As an example, Fig. 7 shows fires detected by MODIS in June-July 2012 and trajectories that passed over them. It is evident that the fires were very close to our site, resulting in strong correlations between $\Delta EC$, $\Delta OC$, and $\Delta CO$ ($R^2>0.9$) (Fig. 8). The $\Delta OC/\Delta EC$ and $\Delta EC/\Delta CO$ ratios from the fire pollution episodes are $26.2 \pm 0.1$ and $1.4 \pm 0.1$ ng m$^{-3}$ ppb$^{-1}$, respectively (Table 3). As discussed in Section 4.5 and shown in Table 3, these values are in the range of emission ratios found in boreal forest fires dominated by smoldering. In addition, the $\Delta TCM/\Delta PM$ ratio was close 1, indicating that carbonaceous material was the dominant component in PM10. The $\Delta WSOC/\Delta OC$ ratio ranged from 0.47 to 0.68 and did not correlate with meteorological conditions (Table 3), probably due to the short-distance transport of the pollution plumes.

### 3.5.3 Polluted air in the transition seasons

In the spring season, strongly elevated concentrations of CO, PM10, and carbonaceous species were observed on 28 April 2010 (Table 4). Figure 9a shows large-scale fires as observed by the MODIS satellite. In this period, the fires were mainly located in the middle belt of European Russia and in the area covering the south and southwest of Siberia and northern Kazakhstan (Fig. 7a). The HYSPLIT 4-day trajectories show that the air masses that arrived at ZOTTO on 28 April 2010 had passed over these fire zones. In particular, the low-altitude air masses that arrived at ZOTTO from 18h to 24h on 28 April had passed over fires located in southern Siberia (Novosibirsk region), whereas the high-altitude airflows that arrived between 00h and 16h of 28 April 2010 had traveled over the fires in the European part of Russia and southwestern Siberia (Tyumen region) (Fig. 9a, c). As a result, the CO, PM10, OC, and EC concentrations reached 261 ppb, 54 μg m$^{-3}$, 18.9 μg m$^{-3}$ and 1.5 μg m$^{-3}$ (Table 4), exceeding their REBS background values by factors of 1.7, 10, 13, and 15, respectively. That was despite the fact that the median trajectory height was well above the mixed layer depth (300 m vs. 74 m) and that precipitation (4.2 mm) could have removed aerosol species (Table 4). It is worth noting in this regard that chemical compounds released by large fires may be injected into the atmosphere up to altitudes of several kilometers, reaching the free atmosphere and even the lower stratosphere (Liousse et al., 1996; Andreae et al., 2001, 2004; Trentmann et al., 2006; Rosenfeld et al., 2007).

The large-scale biomass burning that occurs annually in April - May in southern Russia, including Siberia and northern Kazakhstan, is caused by agricultural fires started by farmers

clearing the fields. Agricultural prescribed burning in Russia is estimated to total 30 million ha annually, of which about 5 million ha is stubble burning (wheat, rye, barley, and oats straw) and 25 million ha are pastures and hayfields (Shvidenko et al., 1995). These burns often escape and cause forest wildfires. The air masses impacted by these fires can be transported over long dis-
5 tances. For example, Siberian and Kazakhstan agricultural emission plumes were sampled by flights over northern Alaska (Warneke et al., 2009; Kondo et al., 2011a) and the Arctic (Matsui et al., 2011).

Our $\Delta OC/\Delta EC$ ratio obtained for agricultural plumes during pollution episodes is 12.7 ± 2.7, which is slightly higher than values reported for prescribed fires during field and chamber
experiments (Table 3). The reason could be chemical and physical ageing processes during long-range transport of polluted air masses from the sources to ZOTTO. Adsorption, condensation and cloud processing of semi-volatile organic compounds tend to increase the OC abundance in the particulate matter (Kanakidou et al., 2005; Hallquist et al., 2009; Sakamoto et al., 2016; Konovalov et al., 2015, 2017). Particularly, Chi et al. (2013) showed that during long-range
transport of agricultural plumes the particles grew from 40 nm to 100 nm within the first 30 h, with a growth rate of about 2 nm h$^{-1}$ due to condensation and coagulation.

The $\Delta EC/\Delta CO$ ratio for the selected agricultural burning episode is 14.3 ± 4.4 ng m$^{-3}$ ppb$^{-1}$, comparable to values reported for prescribed field fires by Pan et al. ( 2012), Zhang et al. (2015), and Li et al. (2007), but higher than those derived during chamber combustion experiments
(Dhammapala et al., 2007b; Hayashi et al., 2014) (Table 3). It should be noted that in combustion experiments the emission factor for different carbon species strongly depends on the type of residue combusted, initial moisture content, burning phase, and analytical instrument used for EC analysis. As a result, different studies yield a wide range of OC/EC and EC/CO ratios, as shown in Table 3 ("Agricultural fires" section). In addition, the long-range transport of agricul-
tural plumes can change these ratios significantly due to aerosol aging and wet scavenging.

As a case in point, the previous long-distance observations of spring agricultural plumes from Siberia and Kazakhstan, sampled by flights from northern Alaska (Fairbanks), reported $\Delta EC/\Delta CO$ values of 10 ± 5 (Warneke et al., 2009) and 8.5 ± 5.4 ng m$^{-3}$ ppb$^{-1}$ (Kondo et al., 2011a) (Table 3). These ratios are almost a factor of 1.5 lower than our ratio of 14.3 ± 4.4 ng m$^{-3}$
30 ppb$^{-1}$, although the EC values from the SP2 and the Sunset (TOT) instruments are directly comparable (Kondo et al., 2011b); e.g., Laborde et al. (2013) reported a slope between the two methods of 1.05. It is reasonable therefore to explain this difference with different air mass history (i.e., wind patterns, boundary layer dynamics, and precipitation), while of course differences in combustion conditions cannot be excluded either. The distance between the agricultural fires and

the sampling areas is almost five times greater for Fairbanks than for ZOTTO, and precipitation could have substantially decreased the EC concentration at Fairbanks as compared to ZOTTO.

In turn, our ratio is almost two times lower than the values of 30 ng m$^{-3}$ ppb$^{-1}$ measured by Cristofanelli et al. (2013) at the Mt. Cimone remote station (Italy) and 22 ng m$^{-3}$ ppb$^{-1}$ obtained at the ZOTTO site by Chi et al. (2013) (Table 3). Besides differences in the air mass history, in this case the observed disagreement could have been caused by the different methods used for EC analysis. The MAAP (multi-angle absorption photometer) and PSAP instruments used by Cristofanelli et al. (2013) and Chi et al. (2013), respectively, utilize light attenuation on filters. Both methods tend to overestimate the mass concentration of EC because of the lens effects of organic matter and the light-absorbing brown carbon substances deposited on the filter matrix (Andreae and Gelencsér, 2006; Bond et al., 2006; Reisinger et al., 2008; Lack et al., 2010). In contrast, the thermo-optical NIOSH 5040 method (used in this study) tends to underestimate EC, especially for biomass aerosols, due to its potential evaporation in the He atmosphere at temperatures below 750°C and/or the catalytic effects of potassium salts (Chow et al., 2001). For example, an intercomparison of EC measurements in central eastern China indicated that the EC mass measured by MAAP was systematically 45–54% higher than the EC mass determined by the thermal-optical transmittance method (Kanaya et al., 2008).

Figure 10 shows the scatterplot between PSAP-derived BC$_e$ and thermal-optical EC obtained in this study during 19 April 2010 - 15 May 2012. The BC$_e$ concentration was calculated from Eq. (1) with the commonly used value of $\alpha_{abs}$ of 10 m$^2$ g$^{-1}$. A good correlation between EC and BC$_e$ for the entire period with R$^2$ = 0.90 is observed, with a BC$_e$/EC slope of 1.67 ± 0.05. Thus, our ΔEC/ΔCO ratio of 14.3 ± 4.4 ng m$^{-3}$ ppb$^{-1}$ obtained for agricultural fires (Table 4) can be converted to a ΔBC$_e$/ΔCO ratio of 24 ± 7.3 ng m$^{-3}$ ppb$^{-1}$. This value is within the range of 22 ng m$^{-3}$ ppb$^{-1}$ and 30 ng m$^{-3}$ ppb$^{-1}$ reported by Chi et al. (2013) and Cristofanelli et al. (2013), respectively, using filter-based light absorption techniques (Table 3).

In the fall season, elevated concentrations of CO, PM10, and carbonaceous species were observed during 14 - 15 October 2010. Figure 9b shows that air masses arriving at ZOTTO were impacted by large-scale fires and anthropogenic emissions from industrialized urban sites. As in the case of the spring pollution episodes, the observed fires are caused by agricultural burning of crop, pasture, and hayfield residues. Figure 9d shows that about half of the 4-day HYSPLIT trajectories stay well below 300 m, that is, the air masses moved mainly within the boundary layer (median MLD: 272 m, interquartile range, 326 m) implying that dilution for these pollution plumes was modest. In addition, the precipitation amounts along the trajectories were small (Table 4). For the selected period, the ΔOC/ΔEC ratio is 18.3 ± 4.0, falling between the values ob-

tained during summer burning (24 - 35) and winter pollutions (4.3 – 5.7) episodes and indicating that the polluted air masses were enriched by organic species from biomass burning (Table 4), but with more of a smoldering component than in the spring episode. In contrast, the $\Delta TCM/\Delta PM$ (=0.45) was lower than that in spring and summer (Table 4), suggesting that there was a strong contribution of inorganic compounds from urban emissions to the PM10. The $\Delta EC/\Delta CO$ ratio for the selected period was $6.0 \pm 2.0$ ng m$^{-3}$ ppb$^{-1}$, which lies in the middle of the range of values from the other seasons and therefore also suggests a mixture of sources (Tables 3 and 4), including a significant impact of anthropogenic emissions on the air mass composition arriving at ZOTTO during the selected period.

Summing up the analysis of seasonal and episodic pollution presented above we can conclude the following. In the winter season pollution originates mainly from the big cities to the south and southwest of the ZOTTO site. In summertime, strong Siberian forest fires are responsible for elevated concentrations of CO, PM, and carbonaceous species, and in the spring and fall seasons both agricultural burning and anthropogenic emissions contribute to the observed elevated concentrations at ZOTTO. In general, our results are consistent with the earlier analysis of the pollution sources conducted for the ZOTTO station during the 2007 and 2008 warm seasons (Vasileva et al., 2011), between September 2006 and December 2011 (Chi et al., 2013), and from September 2006 to December 2011 (Heintzenberg et al., 2013).

**3.6 Characterization of background air**

Figure 11 shows the time series of CO, EC, OC, and PM10 background concentrations, which were obtained based on the REBS algorithm. The median background concentrations (25$^{th}$ -75$^{th}$ quartiles) across the entire study period for EC, OC, and PM10 were 0.06 (0.04-0.08) µg m$^{-3}$, 0.79 (0.58-1.29) µg m$^{-3}$, and 4.4 (3.2-7.5) µg m$^{-3}$, respectively (Fig. 11c,d).

Both CO (Fig. 11a) and EC (Fig. 11b) concentrations show synchronous seasonal variations with a clear annual period of 365±4 days (sine fit). The maximum concentrations of CO and EC fall in the winter at ~ 160 ppb and ~ 0.1 µg m$^{-3}$, respectively, while the minimal values of ~ 100 ppb and ~0.03 µg m$^{-3}$ are observed in the summer season. The Pearson correlation coefficient between background EC and CO concentrations over all seasons is 0.59, which is a fairly close relationship considering that both species come to ZOTTO from remote combustion sources and that atmospheric processes exert different influences on the two species, as discussed above.

In contrast, the OC (Fig. 11c) and PM10 (Fig. 11d) concentrations do not exhibit clearly pronounced annual variations. Such a lack of seasonality of the total aerosol concentration had

already been noted by Chi et al. (2013) based on measurements of aerosol volume concentration and scattering coefficients. This is likely the result of a combination of effects, where regional production of biogenic SOA from VOCs in summer is balanced by higher inputs of pollution aerosols in winter. As noted above, aerosol aging and precipitation could further modify the proportion between different species. It is known that aged organic species and water soluble inorganic compounds are removed from the atmosphere by wet deposition more efficiently than EC (Cerqueira et al., 2010; Witkowska et al., 2016). In addition, during the transition from winter to summer, differential scavenging of the aerosol species can occur in mixed-phase clouds due to the Wegener-Bergeron-Findeisen effect (Wegener, 1911; Bergeron, 1935; Findeisen, 1938). In this type of clouds, the scavenging efficiency of EC decreased more than a factor of five (Cozic et al., 2007; Qi et al., 2017).

### 3.6.1 Near-pristine air

Over the entire measurement period, the cleanest air masses were observed in summertime, as long as biomass-burning pollution did not affect the site. We selected three clean periods based on the selection criterion that EC concentrations had to be below the detection limit of $0.02 \pm 0.01$ $\mu gC$ $m^{-3}$. Table 5 summarizes the dates and chemical species concentrations obtained during these time periods. The lowest concentrations were observed from 7 July to 1 August 2010. For this period, the EC concentration was below the detection limit and the CO mixing ratio was as low as $93 \pm 11$ppb (Table 5). Backward trajectories computed for this time period together with fire maps (Fig. 12a, b) show that the trajectories either did not pass through the fires (panels a.2, a.6, a.8, a9, a.10) or just barely touched them (Fig. 12a; panels a.3 - a.5, a.7, a.11- a.14). Given the low CO concentration, a measureable pollution input from the fires is not to be expected. In addition, any biomass burning aerosol that may have been transported in the boundary layer (Table 5, column 1 and 2) could have been effectively removed via precipitation (APT = 7.3 mm) (Kanaya et al., 2016; Ohata et al., 2016). Figure 12c shows that precipitation with different intensity occurred along the air mass trajectories before they reached the measuring site. As a result the PM10 and OC concentrations were as low as $0.77 \pm 0.09$ and $0.29 \pm 0.02$ $\mu g$ $m^{-3}$, respectively. The TCM/PM ratio was the highest among the clean periods (Table 5), suggesting that biogenic SOA was the main component in the particulate matter.

For the other relatively clean periods, i.e., 6 - 10 July 2011 and 15 - 27 June 2013, the PM10 and carbonaceous species concentrations were much higher than those presented above (Table 5). Most likely, the air masses arriving at ZOTTO during these time intervals were partly influenced by remote pollution sources. Figures 13a,b show that the location of some hot spots

(bold red circles) coincides with gas and oil field production, suggesting that these hot spots represent gas flaring (Elvidge et al., 2011; Anejionu et al., 2015), which is a significant source of EC and other carbonaceous species (Stohl et al., 2013). Therefore, the air masses coming over the land surface from the northwest (Fig. 13c, d) could have accumulated gas flaring products emitted from oil production sites in the West Siberia Basin. In addition, the southeastern air masses arriving at ZOTTO on 19 June 2013 (Fig.13b) passed over forest fires and most likely were also enriched with biomass combustion products. On the other hand, for some time intervals, precipitation along the air mass trajectories was light and scattered (Fig. 13e, f) indicating that aerosol scavenging by precipitation was modest. Thus, during otherwise clean periods, the presence of remote pollution sources (gas flaring) as well as different meteorological conditions (precipitation) could lead to very different PM10 (0.77 vs. 8.7 and 5.3 $\mu$g m$^{-3}$) and OC (0.29 vs. 1.14 and 1.46 $\mu$g m$^{-3}$) concentrations (Table 5).

In the growing season, forested boreal regions emit BVOCs (isoprene, terpenes and sesquiterpenes) into the atmosphere (Kourtchev et al., 2005; Ehn et al., 2012; Noe et al., 2012; Ebben et al., 2011; Vestenius et al., 2014) and their oxidation products contribute to SOA formation or condense onto already formed particles during transport, increasing aerosol sizes and mass (Tunved et al., 2006; Hallquist et al., 2009; Chi et al., 2013; Liao et al., 2014). Since the emission of the organic particle precursors depends on air temperature (Isidorov et al., 1985; Lappalainen et al., 2009), the SOA mass concentration also positively correlates with temperature (Leaitch et al., 2011; Corrigan et al., 2013; Liao et al., 2014). Earlier measurements conducted at ZOTTO in 2009 during a pristine summer period had shown a clear temperature dependence with increasing number concentration and particle size for higher air temperatures. The particle distribution during this period was dominated by an elevated size mode at 160–200 nm and the aerosol volume increased by a factor of three as the temperature increased from 10 to 30 °C (Chi et al., 2013).

In this study, we use the filter-based PM10 and OC concentrations obtained during the near-pristine summer periods in 2010-2014 (14 time intervals, mostly of 3-11 days) together with average ground-level temperatures estimated from the NCEP/NCAR Reanalysis meteorological profiles along each trajectory for an analysis of the temperature dependence of aerosol concentrations. This is in contrast to the previous study (Chi et al., 2013), where aerosol particle size spectra and surface temperature were measured simultaneously at the ZOTTO site over 1-hour intervals. For the analysis presented here we used trajectories initialized at 2 m height, because the temperatures for these trajectories were representative of the surface temperature. A comparison of trajectory end points for 300 m and 2 m starting height found similar horizontal transport patterns, indicating that both trajectories represent horizontal flow in the boundary lay-

er. The scatter plots of PM10 and OC vs. temperature (Figs. 14a and 14b) show no correlation. Most likely the long-range transport of aerosol particles to the ZOTTO site and variable meteorological conditions (wind direction, precipitation) obscured any effect of temperature variations on the particle mass concentration. However, Fig. 15 shows that the ratio OC/PM, which eliminates the effects of transport and removal processes that affect the absolute mass concentration, shows a strong dependence on temperature. There is a clear exponential temperature dependence of the OC/PM ratio, with $R^2$=0.66, which follows the exponential increase of the monoterpene emission rate observed for Scots pine (Isidorov et al., 1985), the dominant tree species in the West Siberia Lowland.

Our average summer background concentrations can be compared with those measured in the Amazon Basin (Brazil) at FNS (Fazenda Nossa Senhora Apareçida) during the wet period from 30 October to 14 November 2002 (Decesari et al., 2006) and at ATTO (Amazon Tall Tower Observatory) in the wet season, i.e. February – June 2014 (Andreae et al., 2015). The FNS site is located in a rural area in the state of Rondônia Brazil (10° 45′ S; 62° 21′ W, 315 m a.s.l.) and the ATTO site is in the Uatumã Sustainable Development Reserve (USDR) in the central Amazon  (02° 08' S; 59° 00' W). Complete descriptions of the FNS and ATTO sites are given by Andreae et al. (2002; 2015). For the FNS samples the same NIOSH 5040 temperature protocol was used for OC/EC analysis, while at ATTO an Aerosol Chemical Speciation Monitor (ACSM) (Ng et al., 2011) and a filter-based MAAP were used for OC and $BC_e$ determination, respectively. Accordingly, the $BC_e$/EC ratio of 1.67 ± 0.05 obtained in this study was applied to convert the $BC_e$-derived concentrations at ATTO into EC values.  Table 6 summarizes the carbonaceous species concentrations obtained at the Amazonian sites and at ZOTTO. It is seen that in spite of the strong difference in climatic zones and vegetation type, the EC, OC, and WSOC concentrations as well as their ratios are in fairly good agreement.  Particularly, the average concentrations of OC and the range of their variations are very close.  This is likely the result of a combination of effects, where production of biogenic SOA from VOCs is balanced by precipitation removal. In addition, aging processes, such as oxidation by ozone or hydroxyl radical, oligomerization, and condensation may largely offset the impact of the vegetation type on the SOA composition. A consequence of the atmospheric evolution of SOA is the convergence of its chemical composition and physical properties (Andreae, 2009). Particularly, the CCN and hygroscopic properties of the SOA formed in the rain forest and boreal forest environments become very similar (Mikhailov et al., 2015b; Pöhlker et al., 2016).

**Conclusion**

We presented time series of CO, PM10, and carbonaceous species (EC, OC, WSOC) concentrations measured at the ZOTTO site in central Siberia from April 2010 to June 2014. Using a non-parametric statistical approach, we separated the time intervals representing background and polluted air. Further, based on the elemental carbon detection limit we selected near-pristine periods. We used the HYSPLIT back trajectory model coupled with the NCEP/NCAR meteorological dataset to trace the air mass trajectories for selected time periods and to estimate meteorological parameters along the trajectories. In addition, the satellite MODIS instrument was used for monitoring of fires.

In the summer season, when CO mixing ratio, PM10, and carbonaceous species concentrations reach their maxima, aerosol particles from large-scale boreal wildfires are dominant. Overall, 48% of air masses in the summer season are classified as polluted, with pollutants originating mostly from nearby forest fires. These plumes are associated with extremely high seasonal concentrations of CO (667 ± 711 ppb), PM10 (59.4 ± 53.1 μg m$^{-3}$), and OC (26.1 ± 26.7 μg m$^{-3}$), with ΔOC/ΔEC and ΔEC/ΔCO ratios of 26.2 ± 0.1 and 1.3 ± 0.1 ng m$^{-3}$ ppb$^{-1}$, respectively.

In winter, polluted air masses account for 75% of days at ZOTTO. The HYSPLIT trajectory analysis shows that pollution plumes originate mainly from the big industrialized cities to the south and southwest of the site. During the winter pollution events the ΔOC/ΔEC and ΔEC/ΔCO ratios are 3.9 ± 0.6 and 5.8 ± 0.7  ng m$^{-3}$ ppb$^{-1}$, respectively, suggesting that the contribution of coal and fossil burning for heating was dominant.

Our analysis of the spring aerosol samples indicates that elevated CO and aerosol species concentrations associated with long-range transport of pollution plumes from the steppe zone of southern Siberia and northern Kazakhstan result from agricultural fires. For one extreme pollution episode observed on 28 April 2010, the CO, PM10, EC, and OC concentrations were as high as 261 ± 12 ppb, 54.4 ± 3.7, 1.5 ± 0.3, and 18.9 ± 1.2 μg m$^{-3}$, respectively, with ΔOC/ΔEC = 12.7 ± 2.7 and ΔEC/ΔCO = 14.3 ± 4.4 ng m$^{-3}$ ppb$^{-1}$. Overall, the frequency of polluted episodes in the spring season was estimated to be 42%.

In fall, the most polluted air masses arriving at ZOTTO were caused by anthropogenic emissions from southern and southwestern urban and industrial sites enriched by combustion products from scattered agricultural fires with ΔOC/ΔEC ratios of 18.3 ± 4.0 and ΔEC/ΔCO of 6.0 ± 2.0 ng m$^{-3}$ ppb$^{-1}$. Over the entire fall season, 42% air masses were classified as polluted. Based on the REBS selection algorithm, the frequency of background conditions by season is 25%, 58%, 39%, and 58% for winter, spring, summer, and fall, respectively. The EC and CO background concentrations show a sine-like behavior with maximum values in winter of 151± 20 ppb and 0.08 ± 0.03 μg m$^{-3}$ and minimum values in summer of 114 ± 15 ppb and 0.03 ± 0.02 μg

m$^{-3}$, respectively. The observed background concentrations are mostly due to different degrees of dilution and removal of pollution in the air masses arriving at the ZOTTO site from remote sources.

Our analysis of the near pristine conditions, based on the EC detection limit of $0.02 \pm 0.01$ µgC m$^{-3}$ shows that the longest periods with clean air masses were observed in summer (frequency of 17%). For this period, the variations in the OC/PM ratio closely correlated with those in air temperature, which implies that biogenic sources of OC formation are dominating. Against a low concentration of CO, EC, and OC observed in the near-pristine summer episodes it was possible to identify pollution plumes that most likely arrived at ZOTTO from crude oil produc-

tion sites located in the oil-rich regions of Western Siberia.

Over the entire measurement campaign, the WSOC/OC ratio remains relatively stable and ranged from 0.5 to 0.7, indicating that partitioning between water soluble and water insoluble organic matter is not significantly dependent on the source of the polluted air mass. Most likely, aging processes and precipitation scavenging control the ratio between them.

Further studies at ZOTTO may investigate the link between aerosol and climate: e.g., the influence of rising temperatures on an enhancement of biogenic SOA formation, as well as a quantification of anthropogenic emissions, which produce feedbacks on cloud formation, tropospheric radiation balance, and precipitation patterns (Kulmala et al., 2004; Goetz et al., 2007). The continuing measurements at ZOTTO will help to detect these trends in aerosol emissions

and their influences on future climate.

*Data availability.* Data used in this study can be made available upon request to the author.

*Competing interests.* The authors declare that they have no conflict of interest.

*Acknowledgements.* This work was supported by the Max Planck Society (MPG), RFBR grants 16-05-00718 and 16-05-00717, and Saint Petersburg State University (SPBU) grant BRICS 11.37.220.2016. We thank the Geomodel Research Center at Saint Petersburg State University for help with the chemical analysis of the ambient aerosol samples. We acknowledge Christopher Pöhlker for his generous help during the ZOTTO campaign. This work was partly per-

formed within the research agenda of the Pan-Eurasian Experiment (PEEX) (https://www.atm.helsinki.fi/peex/).

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

**Table 1.** Mean (with ± st. dev., where available) and median (min-max) concentrations (µg m$^{-3}$) of EC, OC, and WSOC, together with EC/TC and WSOC/OC ratios measured at remote background sites compared with ZOTTO data obtained from April 2010 to June 2014.

| Site | Sampling period | EC µg m$^{-3}$ | OC µg m$^{-3}$ | WSOC µg m$^{-3}$ | EC/TC | WSOC/OC | Reference |
|---|---|---|---|---|---|---|---|
| Hyytiälä, Finland | 02.08. - 29. 08.2007 | 0.12 (0.01-0.34)[a] | 1.45 (0.24-3.9) | 0.91 (0.11-2.5) | 0.07 ± 0.02 | 0.62 ±0.09 | Chi, 2009[a] |
| K-puszta, Hungary | 04.06. - 10.07.2003 | 0.19 (0.08-0.59) | 3.7 (1.61-5.7) | 2.1 (0.73-3.8) | 0.06 ± 0.02 | 0.57 ± 0.11 | Chi, 2009[a] |
| K-puszta, Hungary | 24.05. - 29.06.2006 | 0.16 (0.02-0.45) | 2.5 (0.85-5.9) | 1.37 (0.38-3.8) | 0.07 ± 0.03 | 0.55 ± 0.11 | Chi, 2009[a] |
| ZOTTO, Russia | 2010 – 2014, summer | 0.13 (0.02-4.0) | 4.4 (0.3-106) | 3.3 (0.2-45) | 0.03 ± 0.01 | 0.65 ± 0.13 | This study[a] |
| High-altitude remote sites | | | | | | | |
| Mt. Sonnblick, Austria (3160 m) | 10.2003 - 10.2005 | 0.15 | 0.93 | 0.60 | 0.20 | 0.65 | Pio et al., 2007[b] |
| Puy de Dôme, France (1450 m) | 10.2003 - 10.2005 | 0.22 | 1.60 | 1.08 | 0.12 | 0.68 | Pio et al., 2007[b] |
| Schauinsland, Germany (1205 m) | 10.2003 - 10.2005 | 0.29 | 2.39 | 1.86 | 0.11 | 0.78 | Pio et al., 2007[b] |
| Zhuzhang, China (3583 m) | 06.2004 - 05.2005 | 0.34 ± 0.18 | 3.1±0.9 | ND | 0,10 ± 0.05 | ND | Qu et al., 2009[c] |
| Akdala, China (562 m) | 06.2004 - 05.2005 | 0.35 ± 0.31 | 2.9 ± 1.6 | ND | 0.11 ± 0.10 | ND | Qu et al., 2009[c] |
| Qinghai Lake, China (3200 m) | 11.2011 - 11.2012 | 0.84 ± 0.46 | 3.46 ± 1.64 | 1.55 ± 0.77 | 0.20 ± 0.14 | 0.45 ± 0.26 | Zhao et al., 2015[c] |
| ZOTTO, Russia | 04.2010 - 06.2014 | 0.25 ± 0.43 0.12 (0.02-4.0) | 4.66 ± 12.0 1.46 (0.29-106) | 2.69 ± 6.0 0.92 (0.05-45) | 0.08 ± 0.06 | 0.63 ± 0.14 | This study[a] |

Methods, protocol used for EC/OC: [a]Thermal/optical, NIOSH (TOT); [b]Thermal/optical; [c]Thermal/optical, preheating at 600 °C without O$_2$; [c]Thermal/optical, IMPROVE (TOR); TOT (TOR), thermal-optical transmittance (reflectance).

**Table 2.** Seasonal concentration (± st. dev.) of CO, PM10, and carbonaceous species together with ratios and frequency representative of polluted, background, and clean periods obtained at ZOTTO from April 2010 to June 2014.

| Season and conditions | Frequency[a] | | CO[b] | CO | PM10 | EC | OC | WSOC | WSOC/OC | TCM/PM |
|---|---|---|---|---|---|---|---|---|---|---|
| | % | | ppb | ppb | μg m⁻³ | μg m⁻³ | μg m⁻³ | μg m⁻³ | | |
| **Winter** | | | | | | | | | | |
| Polluted | 75 | 47 | 181 ± 24 | 204 ± 36 | 16.6 ± 12.6 | 0.30 ± 0.20 | 2.7 ± 1.6 | 1.3 ± 1.1 | 0.52 ± 0.17 | 0.38 ± 0.24 |
| Background | 25 | 53 | 148 ± 21 | 151 ± 20 | 5.8 ± 1.9 | 0.08 ± 0.03 | 0.94 ± 0.38 | 0.62 ± 0.34 | 0.61 ± 0.11 | 0.33 ± 0.18 |
| Clean | 1 | 23 | 139 ± 14 | No data | 6.5 ± 2.9 | 0.028 ± 0.004 | 0.75 ± 0.38 | 0.46 ± 0.27 | 0.59 ± 0.11 | 0.13 ± 0.07 |
| **Spring** | | | | | | | | | | |
| Polluted | 42 | 19 | 160 ± 32 | 183 ± 24 | 14.3 ± 11.6 | 0.23 ± 0.27 | 3.1 ± 4.1 | 1.4 ± 1.3 | 0.65 ± 0.12 | 0.36 ± 0.16 |
| Background | 58 | 79 | 155 ± 14 | 153 ± 19 | 5.5 ± 2.9 | 0.07 ± 0.03 | 0.97 ± 0.31 | 0.61 ± 0.25 | 0.63 ± 0.13 | 0.36 ± 0.16 |
| Clean | 9 | 36 | 138 ± 12 | 137 ± 27 | 7.4 ± 5.7 | 0.02 ± 0.01 | 0.81 ± 0.05 | 0.45 ± 0.14 | 0.56 ± 0.16 | 0.31 ± 0.26 |
| **Summer** | | | | | | | | | | |
| Polluted | 61 | 20 | 103 ± 41 | 667 ± 711 | 59.4 ± 53.1 | 0.84 ± 0.92 | 26.1 ± 26.7 | 13.9 ± 11.7 | 0.61 ± 0.13 | 0.75 ± 0.17 |
| Background | 39 | 80 | 92 ± 5 | 114 ± 15 | 6.3 ± 3.9 | 0.03 ± 0.02 | 1.24 ± 0.57 | 0.89 ± 0.45 | 0.69 ± 0.12 | 0.41 ± 0.20 |
| Clean | 17 | 32 | 89 ± 14 | 103 ± 12 | 7.4 ± 4.7 | 0.02 ± 0.02 | 1.36 ± 0.56 | 0.90 ± 0.41 | 0.64 ± 0.10 | 0.43 ± 0.23 |
| **Fall** | | | | | | | | | | |
| Polluted | 42 | 16 | 117 ± 16 | 192 ± 61 | 17.6 ± 10.0 | 0.19 ± 0.12 | 3.4 ± 3.3 | 2.2 ± 1.8 | 0.64 ± 0.12 | 0.34 ± 0.18 |
| Background | 58 | 84 | 108 ± 19 | 118 ± 18 | 7.3 ± 4.6 | 0.06 ± 0.02 | 1.32 ± 0.53 | 0.96 ± 0.47 | 0.70 ± 0.10 | 0.43 ± 0.29 |
| Clean | 10 | 28 | 96 ± 4 | 108 ± 6 | 7.8 ± 4.9 | 0.02 ± 0.02 | 1.44 ± 0.42 | 1.00 ± 0.40 | 0.68 ± 0.11 | 0.39 ± 0.26 |

[a] The values in the left and right columns are frequencies obtained in this study and those reported by Chi et al. (2013), respectively. Polluted and background periods were separated using the REBS algorithm and together comprise 100%, while clean periods were selected independently based on EC (this study) or BC$_e$ concentrations (Chi et al., 2013) as indicated in the text.
[b] CO data from September 2006 to December 2011 at 300 m at ZOTTO (Chi et al., 2013).

**Table 3** Comparison of ΔOC/ΔEC and ΔOC/ΔEC at ZOTTO with values obtained from previous data sources. F, S, and M stand for flaming, smoldering, and mixed burning mode, respectively.

| Location and study period | Instrument, method, protocol for EC | Fuel type | ΔOC/ΔEC | ΔEC/ΔCO (ng m$^{-3}$ ppb$^{-1}$) | References |
|---|---|---|---|---|---|
| **Urban, combustion** | | | | | |
| ZOTTO, winter pollution | Thermal/optical, NIOSH(TOT)[a] | Fossil fuel | 3.9 ± 0.6 | 5.8 ± 0.7 | This study |
| Guangzhou, Oct-Nov. 2004 | Thermal/optical, NIOSH(TOT) | Fossil fuel | 3.2 ± 1.9 | 7.9 ± 0.2 | Andreae et al. (2008b) |
| Guangzhou, July 2006 | Thermal/optical, NIOSH(TOT) | Fossil fuel | | 5.4 ± 0.4 | Verma et al. (2010) |
| Taiyuan, Dec. 2005 – Feb. 2006 | Thermal/optical, NIOSH(TOT) | Fossil fuel | 7.0 ± 3.9 | | Meng et al. (2007) |
| Taiyuan, Sept. 2009 - Feb. 2010 | Thermal/optical, IMPROVE(TOR)[a] | Fossil fuel | 5.4 ± 1.5 | | Li et al. (2012) |
| Beijing, Nov. 2005 - Oct. 2006 | Thermal/optical, NIOSH(TOT) | Fossil fuel | | 3.5 – 5.8 | Han et al. (2009) |
| Xi'an, Dec. 2003 - Feb. 2004 | Thermal/optical, IMPROVE (TOR) | Fossil fuel | 3.5 - 5.1 | | Cao et al. (2005) |
| Tokyo, 2003-2005 | Thermal/optical, NIOSH(TOT) | Fossil fuel | | 5.7 ± 0.9 | Kondo et al. (2006) |
| Houston, Dallas, Sept.-Oct., 2006 | SP2[c], laser-induced incandescence | Fossil fuel | | 5.8 ± 1.0 | Spackman et al. (2008) |
| Beijing, Jan.-Feb., 2009 | Thermal/optical, IMPROVE-A | Fossil fuel | 6.1 ± 1.2 | | Cheng et al. (2011) |
| NCP[b], Apr.- Oct. 2010 NCP and Beijing, Apr.- Oct., 2010 | COSMOS[d], Heated-inlet light adsorption | Fossil fuel | | 8.4 ± 0.4 9.5 ± 2.0 | Wang et al. (2011) |
| **Coniferous forest fires** | | | | | |
| ZOTTO, summer fire episodes | Thermal/optical, NIOSH(TOT) | Boreal forest | 26.2 ± 0.1 | 1.3 ± 0.1 | This study |
| Ontario Canada, August 1989, prescribed burning | Thermal/optical | Boreal forest | 10 - 18 (F) 21 - 95 (S) | | Mazurek et al. (1991) |

| Location and study period | Instrument, method, protocol for EC | Fuel type | ΔOC/ΔEC | ΔEC/ΔCO ($ng\ m^{-3}\ ppb^{-1}$) | References |
|---|---|---|---|---|---|
| Canada, 29 June - 10 July 2008 | SP2[c], laser-induced incandescence | Boreal forest | | 1.7 ± 0.1 (S) <br> 3.4 ± 1.6 (F) | Kondo et al., (2011a) |
| Georgia (US), April 2004 prescribed burning | Thermal/optical, NIOSH(TOT) | Pine dominated forest | 15.4 ± 10.3 (F) <br> 15.4 ± 10.4 (S) | | Lee et al. (2005) |
| Alaska, Missoula, Montana, (combustion chamber) | Thermal/optical, IMPROVE-A | Lodgepole pine <br> Black spruce <br> Ponderosa pine <br> Douglas fir <br><br> Average | 25.1 ±51.6 (S) <br> 10.3 ± 9.2 (S) <br> 36.6 ± 72.6 (S) <br> 72.2 ± 156.1 (S) <br><br> 32.3 ± 119.2 (S) | | McMeeking et al. (2009) |
| Siberian biomass (combustion chamber) | Thermal/optical, NIOSH(TOT) | Scots pine wood <br><br> Debris (needles, branches, cones) | 0.3 - 2.3 (F) <br> 126 -181 (S) <br> 0.6 (F); 35 (S) | | Popovicheva et al. (2015) |
| **Agricultural fires** | | | | | |
| ZOTTO, spring pollution events, 28 April 2010 and 22-24 April 2011 | Thermal/optical, NIOSH(TOT) | Crop residues and steppe grasses | 12.7 ± 2.7 | 14.3 ± 4.4 | This study |
| ZOTTO, spring pollution events, 20 - 22 April 2008 | PSAP[e], light absorption | Crop residues and steppe grasses | | 21.8 | Chi et al. (2013) |
| Agricultural plume, transport from southern Russia to Italy (Mt. Cimone, ICO-OV). 1 - 4 May 2009 | MAAP[f], light absorption | Crop residues and steppe grasses | | 29.8 | Cristofanelli et al. (2013) |
| Agricultural plume, transport from Kazakhstan steppe to Alaskan Arctic, 18 April 2008 | SP2[c], laser-induced incandescence | Crop residues and steppe grasses | | 10 ± 5 | Warneke et al. (2009) |
| Agricultural plume, transport from Kazakhstan and Siberia to Alaska, 12 - 13 April, 2008 | SP2[c], laser-induced incandescence | Crop residues and steppe grasses | | 8.5 ± 5.4 (F) | Kondo et al. (2011a) |

| Location and study period | Instrument, method, protocol for EC | Fuel type | ΔOC/ΔEC | ΔEC/ΔCO (ng m$^{-3}$ ppb$^{-1}$) | References |
|---|---|---|---|---|---|
| Agricultural plume at Indo-Gangetic Plain, India (PM0.95) | Thermal/optical, NIOSH(TOT) | Rice straw<br>Wheat straw | 4.6 ± 2.6<br>2.7 ± 1.0 | | Singh et al. (2016) |
| Post-harvest residues (prescribed field burns) | Thermal/optical IMPROVE (TOR) | Crop residues (mainly wheat straw) | 5.9 ± 5.3 (F)<br>12.3 ± 2.0 (S) | 17.4 ± 5.2 (F)<br>11.8 ± 2.4 (S) | Pan et al. (2012) |
| Post-harvest residues (prescribed field burns) | microAeth. AE51[g], light absorption | Wheat straw<br>Rice straw<br>Rapeseed residue | | 14.4 ± 6.4 (M)<br>9.3 ± 1.1 (F)<br>14.0 ± 7.9 (F) | Zhang et al. (2015) |
| Post-harvest residues (prescribed field burns) | Thermal/optical IMPROVE (TOR) | Corn stover<br>Wheat straw | 11.1 ± 5.8 (F)<br>5.5 ± 2.4 (F) | 8.3 ± 6.7 (F)<br>10.3 ± 4.6 (F) | Li et al. (2007) |
| Post-harvest residues (prescribed field burns) | Thermal/optical, NIOSH(TOT) | Meadow-grass straw<br>Wheat straw | 8.2 ± 2.5 (F)<br>10.0 ± 5.1 (F) | 4.4 ± 3.5 (F)<br>4.5 ± 2.4 (F) | Dhammapala et al. (2007a) |
| Post-harvest residues (combustion hood, open burning) | Thermal/optical IMPROVE (TOR) | Rice straw<br>Wheat straw<br>Barley straw | 4.8 - 9.0[h]<br>12.2 – 16.9[h]<br>7.5 – 17.2[h] | 7.6 – 8.4[h]<br>13.1 – 14.3[h]<br>2.2 – 4.3[h] | Hayashi et al. (2014) |
| Post-harvest residues (burn chamber) | Thermal/optical, NIOSH(TOT) | Meadow-grass straw<br>Wheat straw | 10.9 ± 1.5 (F)<br>5.4 ± 4.3 (F) | | Dhammapala et al. (2007b) |

[a] TOT (TOR), thermal-optical transmittance (reflectance)
[b] NCP, North China Plain;
[c] SP2, single particle soot photometer;
[d] COSMOS, continuous soot monitoring system;
[e] PSAP, Particle Soot Absorption Photometer
[f] MAAP, multi-angle absorption photometer;
[g] AE51, micro aethalometer.
[h] The range is due to different moisture content in the dry (11-13% by weight) and moist (20% by weight) residue.

**Table 4.** Seasonal median concentrations (min-max) of CO, PM10, EC, OC, and WSOC together with ΔEC/ΔCO, ΔOC/ΔEC, WSOC/OC, and TCM/PM ratios during pollution events at ZOTTO from April 2010 to June 2014. The first and second columns are medians (interquartile ranges) of the mixed layer depth (MLD) and air mass trajectory height above ground level (AGL), respectively. The third column is the average of the accumulated precipitation along trajectory (APT) for the period in question. Each $\Delta X_i$ denotes concentration of $X_i$ above background value. The number of aerosol samples is indicated in square brackets.

| Sampling period | MLD | AGL | APT | CO | PM10 | EC | OC | ΔEC/ΔCO | ΔOC/ΔEC | ΔWSOC/ΔOC | ΔTCM/ΔPM |
|---|---|---|---|---|---|---|---|---|---|---|---|
| | m | m | mm | ppb | μg m⁻³ | μg m⁻³ | μg m⁻³ | ng m⁻³ ppb⁻¹ | | | |
| Winter | | | | | | | | | | | |
| 25.12.2010 – 30.12.2011 [5] | 125 (139) | 97 (270) | 3.7 | 218 (205-249) | 14.8 (8-17) | 0.63 (0.4-0.9) | 2.8 (2-5) | 5.9 (4.2-7.1) | 4.3 (1.6-5.2) | ND[a] | 0.35 (0.2-0.9) |
| 10.01.2011 – 15.01.2011 [6] | 73 (75) | 20 (157) | 1.5 | ND[a] | 16.4 (13-20) | 0.56 (0.3-0.8) | 3.7 (3-7) | ND[a] | 4.6 (3.7-6.6) | 0.67 (0.5-0.7) | 0.37 (0.3-0.8) |
| 27.12.2011 – 09.01.2012 [8] | 115 (125) | 225 (574) | 2.9 | 246 (188-263) | 27.3 (10-61) | 0.37 (0.1-0.6) | 2.4 (1-3) | 4.6 (3.1-7.1) | 5.7 (4.0-7.6) | 0.49 (0.2-0.6) | 0.17 (0.1-0.4) |
| Spring | | | | | | | | | | | |
| 28.04.2010 [1]b | 74 (98) | 300 (1180) | 4.2 | 261 ± 12 | 54.4 ± 3.7 | 1.5 ± 0.3 | 18.9 ± 1.2 | 14.3 ± 4.4 | 12.7 ± 2.7 | ND[a] | 0.63 ± 0.08 |
| Summer | | | | | | | | | | | |
| 22.06.2012 – 24.06.2012 [5] | 640 (766) | 452 (378) | 0.3 | 566 (256-838) | 68.0 (11-117) | 1.2 (0.2-1.9) | 32.9 (5-50) | 2.7 (0.9-3.6) | 34.6 (21.4-49.6) | 0.57 (0.5-0.7) | 0.95 (0.8-1.1) |
| 06.07.2012 – 07.07.2012 [1]b | 561 (665) | 92 (81) | 7.1 | 2094 ± 84 | 203.4 ± 4.0 | 4.0 ± 0.3 | 10 6 ± 5.3 | 1.9 ± 0.2 | 26.5 ± 2.1 | 0.55 ± 0.11 | 0.94 ± 0.16 |
| 25.07.2012 – 31.07.2012 [6] | 838 (1162) | 317 (166) | 0.3 | 2093 (330-2278) | 141 (41-154) | 2.1 (0.5-2.1) | 71.9 (18-82) | 1.4 (1.2-2.0) | 27.3 (7.8-52.7) | 0.47 (0.4-0.6) | 0.99 (0.9-1.0) |
| 26.07.2013 – 17.08.2013 [7] | 529 (827) | 358 (384) | 3.1 | 440 (197-801) | 22.1 (10-60) | 0.66 (0.2-1.0) | 11.1 (4-25) | 1.5 (0.7-1.6) | 24.3 (15.2-41.9) | 0.68 (0.6-0.8) | 1.01 (0.8-1.1) |
| Fall | | | | | | | | | | | |
| 14.10.2010 – 15.10.2010 [1]b | 272 (326) | 312 (605) | 0.8 | 230 ± 10 | 34.2 ± 1.8 | 0.64 ± 0.14 | 12.1 ± 0.7 | 6.0 ± 2.0 | 18.3 ± 4.0 | 0.45 ± 0.05 | 0.45 ± 0.08 |

[a]ND – stands for No Data

[b]Only one aerosol sample was collected, therefore the average concentration of CO, PM10, and carbonaceous species as well as their experimental uncertainties are shown.

**Table 5.** Average CO, PM10, and carbonaceous species concentrations (± st. dev.) together with WSOC/OC and TCM/PM ratios obtained during the cleanest periods in the growing season. MLD and AGL are median and (25[th] - 75[th]) quartiles of the mixed layer depth and air mass trajectory height above ground level, respectively.

| Sampling period | MLD | AGL | CO | PM10 | EC | OC | WSOC | WSOC/OC | TCM/PM |
|---|---|---|---|---|---|---|---|---|---|
| | m | m | ppb | µg m⁻³ | µg m⁻³ | µg m⁻³ | µg m⁻³ | | |
| 07.07.2010 – 01.08.2010 | 526 (275 - 831) | 290 (201- 402) | 93 ± 11 | 0.77 ± 0.09 | <0.002 | 0.29 ± 0.02 | 0.16 ± 0.02 | 0.53 ± 0.07 | 0.68 ± 0.10 |
| 06.07.2011 – 10.07.2011 | 747 (367 - 119) | 122 (24 - 241) | ND[a] | 8.7 ± 0.5 | 0.018 ± 0.031 | 1.14 ± 0.05 | 0.61 ± 0.08 | 0.54 ± 0.08 | 0.24 ± 0.02 |
| 15.06.2013 – 27.06.2013 | 415 (208 - 736) | 267 (179 - 337) | 119 ± 14 | 5.3 ± 0.2 | 0.018 ± 0.010 | 1.46 ± 0.08 | 1.22 ± 0.14 | 0.84 ± 0.11 | 0.50 ± 0.03 |

**Table 6** Average background concentrations of carbonaceous species (± st. dev.) or (min – max) as well as their ratios obtained at Fazenda Nossa Senhora (FNS, Rondônia, Brazil) and at the ATTO site (Amazonas, Brazil) during the wet period and those measured in summertime at ZOTTO.

| Site | Sampling period | EC | OC | WSOC | EC/TC | WSOC/OC |
|---|---|---|---|---|---|---|
| | | µg m⁻³ | µg m⁻³ | µg m⁻³ | | |
| ZOTTO, Russia | 2010 - 2014 summer | 0.03 (0.02 - 0.06) 0.03±0.02 | 1.2 (0.3 - 2.2) | 0.88 (0.15-1.6) | 0.03± 0.02 | 0.69 ± 0.12 |
| FNS, Brazil | 30.10. - 14.11.2002, wet period | 0.05 ± 0.03 | 1.4 (0.9 - 2.5) | 0.9 (0.7 - 2.0) | 0.03 ± 0.02 | 0.60 ± 0.12 |
| ATTO, Brazil | February-June, 2014, wet season | 0.04 (0.01 - 0.06) | 1.0 (0.5 - 2.1) | ND[a] | 0.05 (0.02 –0.09) | ND[a] |

[a]ND – stands for No Data

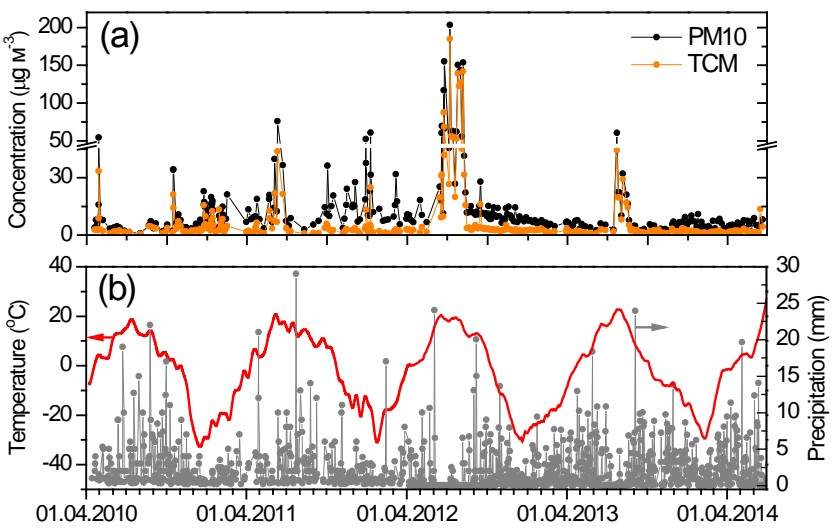

**Figure 1.** Seasonal and interannual variations of the aerosol particulate matter (PM10) and total carbonaceous matter (TCM) concentrations **(a)** and of daily-averaged meteorological parameters **(b)**: temperature - red line; precipitations – gray line and symbols.

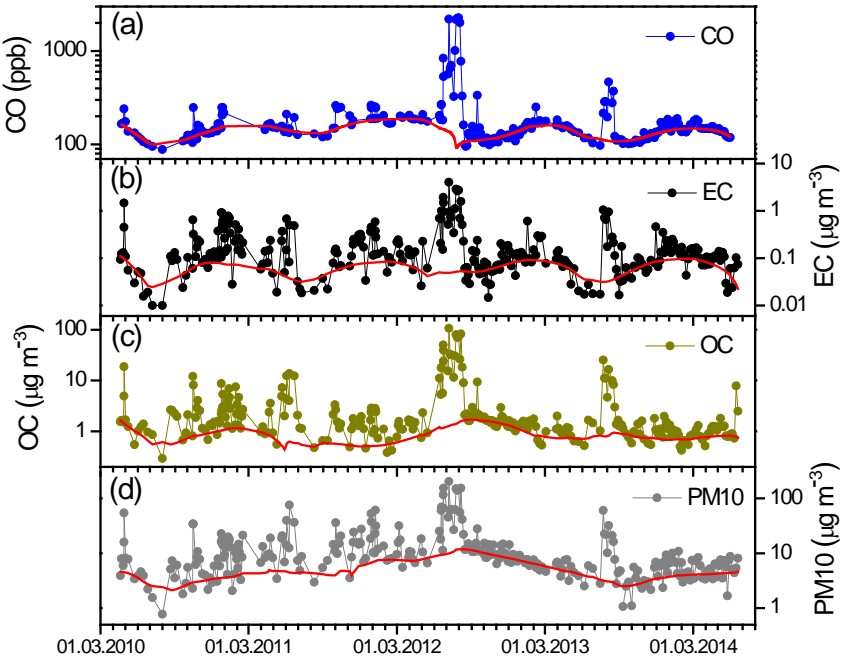

**Figure 2.** Time series of CO **(a)**, EC **(b)**, OC **(c)**, and PM10 **(d)** concentrations at ZOTTO and respective background concentrations (red solid line) as identified by the REBS algorithm (Ruckstuhl at al., 2012).

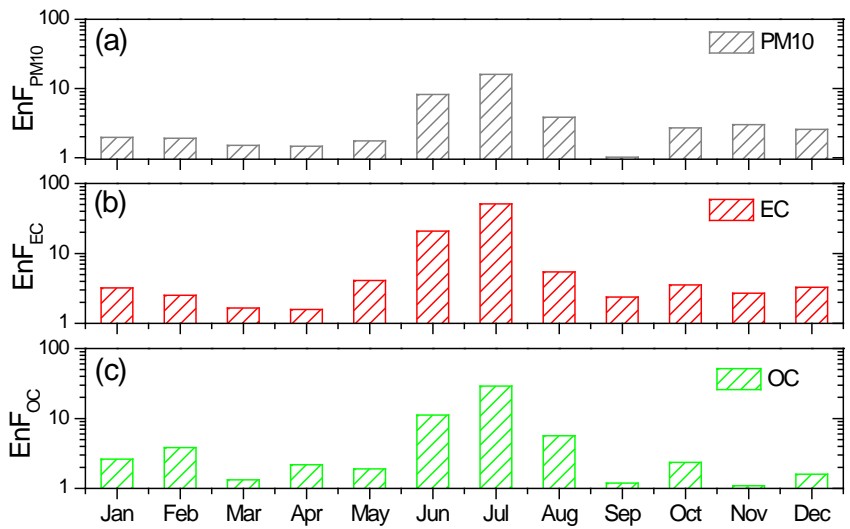

**Figure 3.** Monthly variation of the enhancement factor (EnF) of PM10 (**a**), EC (**b**), and OC (**c**) at ZOTTO from April 2010 to June 2014.

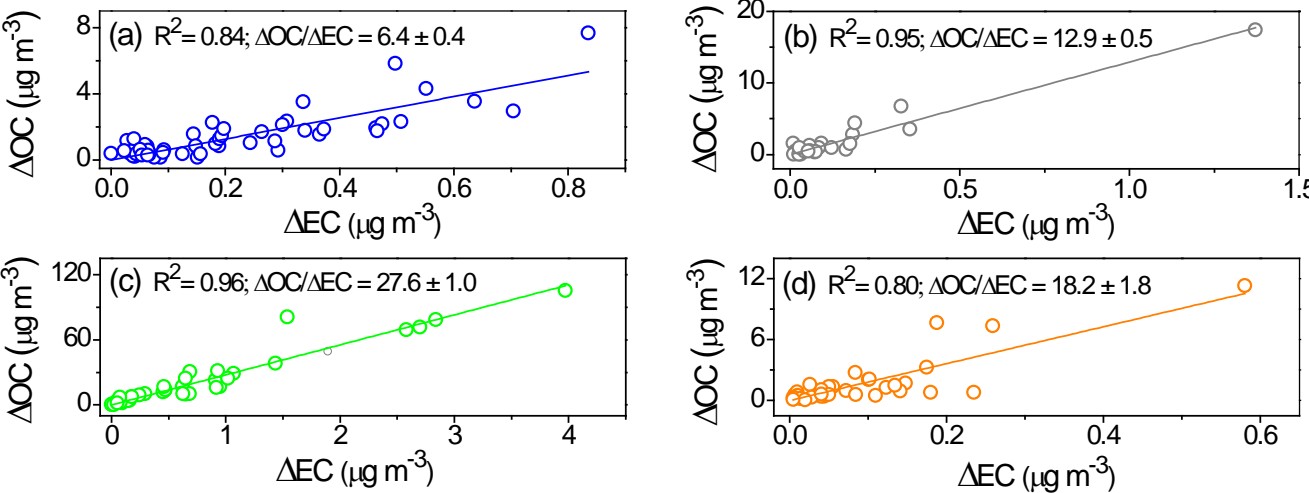

**Figure 4.** Seasonal relationship between EC and OC: winter (**a**), spring (**b**), summer (**c**), and fall (**d**) during pollution periods, using all data obtained at ZOTTO from April 2010 to June 2014. The linear fit is shown for reference: $R^2$ - coefficient of determination; $\Delta OC/\Delta EC$ – slope ±1 st. dev. The polluted $\Delta EC$ and $\Delta OC$ concentrations were calculated as the difference between measured and REBS-defined background concentrations.

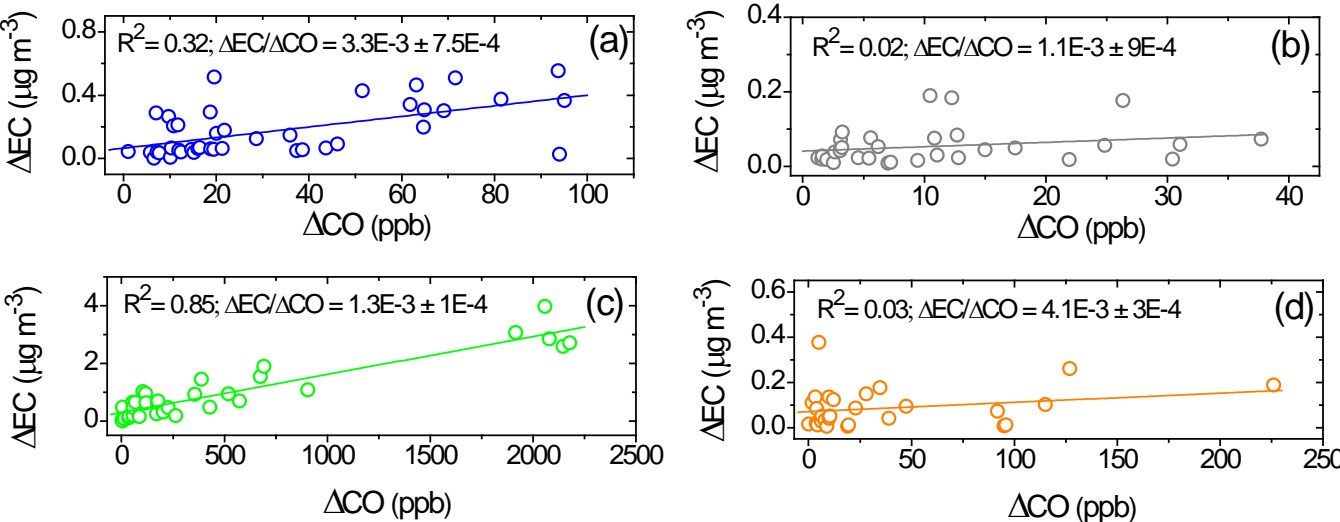

**Figure 5.** Seasonal relationship between ΔEC and ΔCO: winter **(a)**, spring **(b)**, summer **(c)**, and fall **(d)** during pollution periods, using all data obtained at ZOTTO from April 2010 to June 2014. The linear fit is shown for reference: $R^2$ - coefficient of determination; ΔOC/ΔEC– slope ±1 st. dev in µg m$^{-3}$ ppb$^{-1}$. The polluted ΔEC and ΔOC concentrations were calculated as the difference between measured and REBS-defined background concentrations.

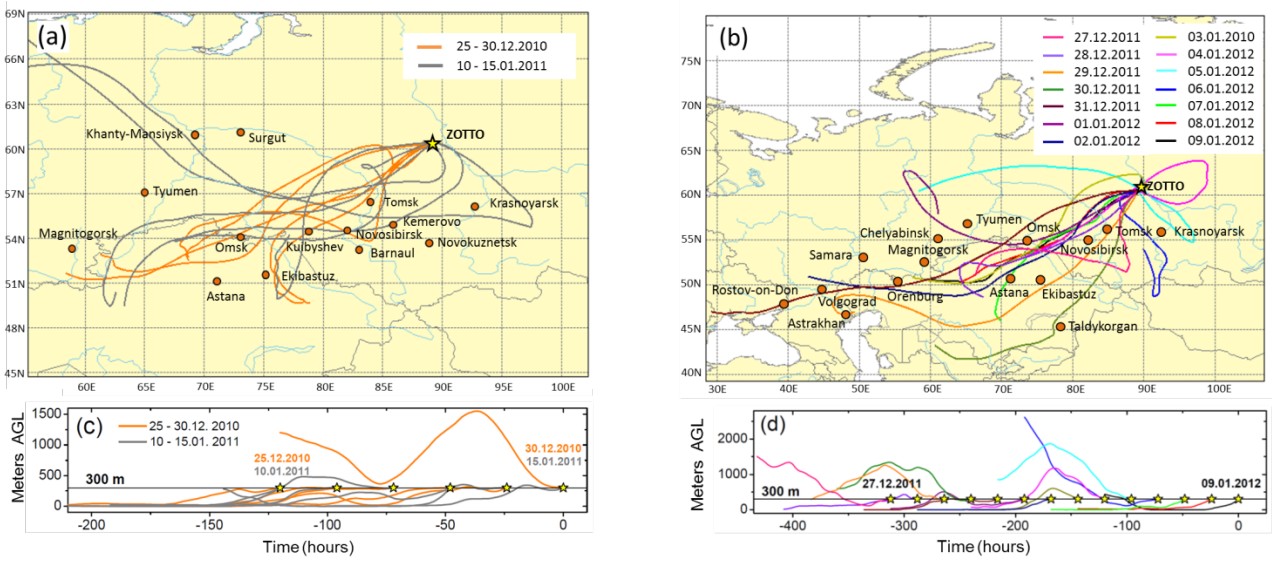

**Figure 6.** 120-h HYSPLIT backward air mass trajectories with 24-h intervals during winter pollution events at ZOTTO at 300 m observed 25 to 30 December 2010 (**a**, orange), 10 to 15 January 2011 (**a**, grey), and 27 December 2011 to 9 January 2012 (**b**), and trajectory height above ground level (AGL) (**c, d**). Stars and dates show the arrival time of the air mass.

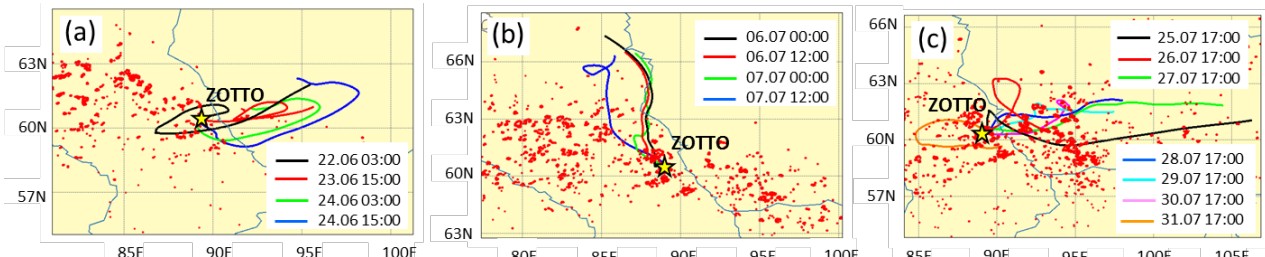

**Figure 7.** 72-h HYSPLIT air mass trajectories during summer biomass burning events at ZOTTO at 300 m: 22 to 24 June 2012 (**a**), 6 to 7 July 2012 (**b**), and 25 to 31 July 2012 (**c**), respectively. Red dots indicate the fire locations.

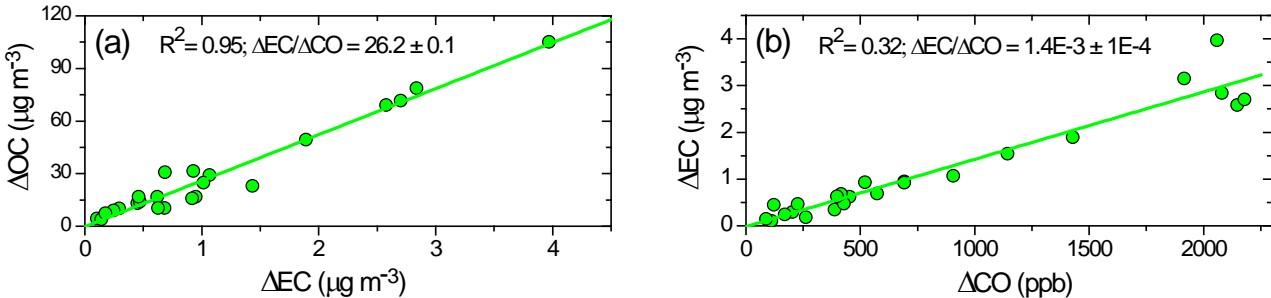

**Figure 8.** Scatter plot and linear regression of ΔEC - ΔOC (**a**) and ΔCO - ΔEC (**b**) during biomass burning episodes in the summer season (Table 2). The ΔEC/ΔCO slope is in µg m$^{-3}$ ppb$^{-1}$.

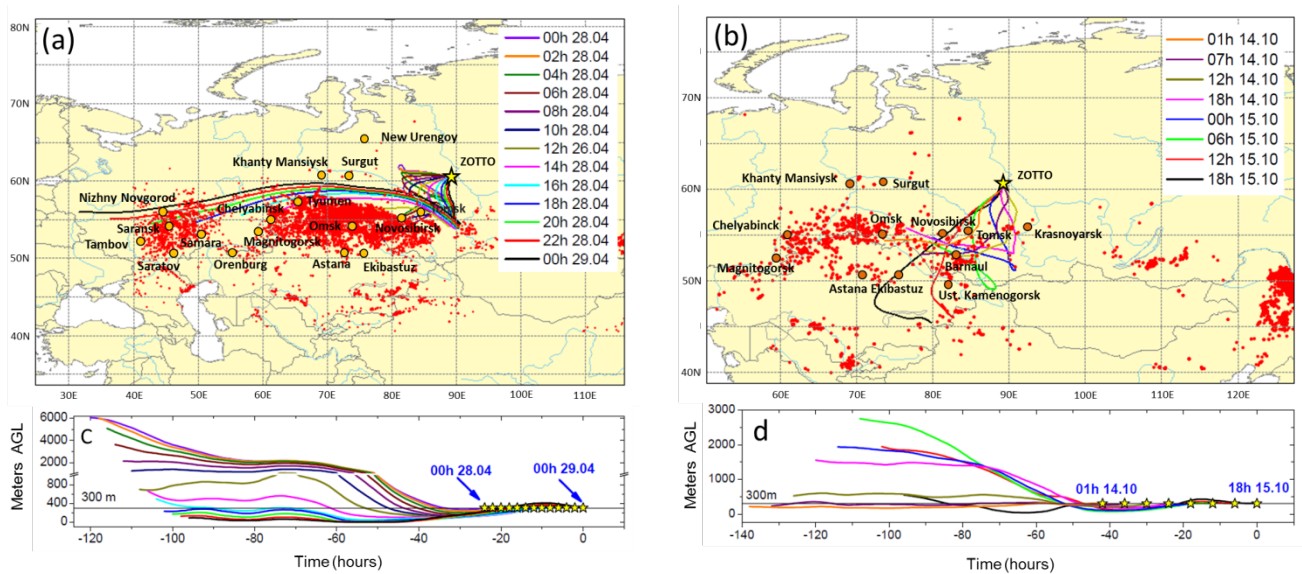

**Figure.9.** 96-h HYSPLIT air mass trajectories during spring pollution events at ZOTTO at 300 m on 28 April, 2011 (**a**) and 14 to 15 October, 2010 (**b**), and their height above ground level (AGL) (**c**) and (**d**), respectively. Red dots indicate the fire locations (**a,b**). Stars and dates show the arrival time of air mass with 2-h (**a,c**) and 6-h (**b,d**) intervals, respectively.

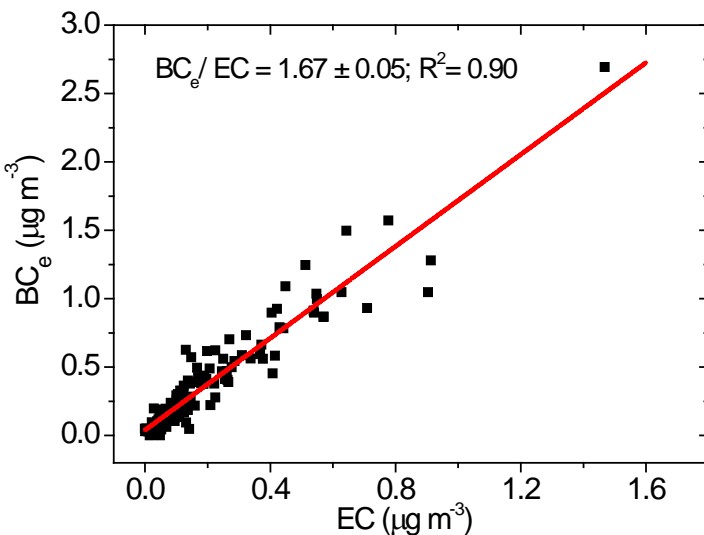

**Figure.10.** Scatter plot and linear regression between thermal-optical EC and light absorption $BC_e$ observed during 19 April 2010 - 15 May 2012. The $BC_e$/EC slope is $1.67 \pm 0.05$ with $R^2 = 0.90$.

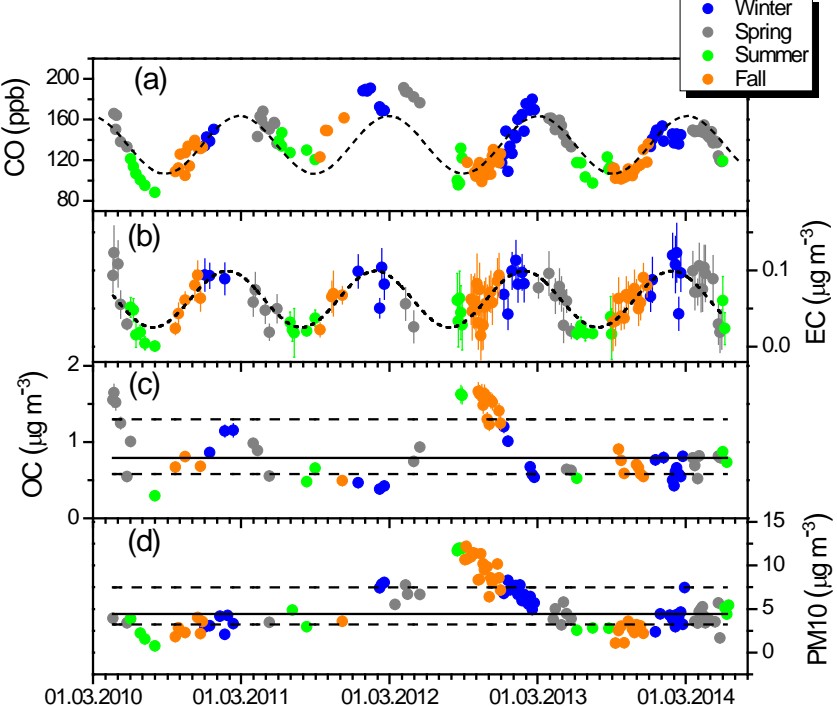

**Figure 11.** Time series of CO **(a)**, EC **(b)**, OC **(c),** and PM10 **(d)** background concentrations for 300 m at ZOTTO. Fit lines: **(a), (b)** sine fit with time period of $365\pm4$ days and coefficient of determination, $R^2 = 0.60$ and $0.58$ respectively; **(c), (d)** median – black solid; 75[th] and 25[th] percentiles – dashed line. Different colors represent the different seasons.

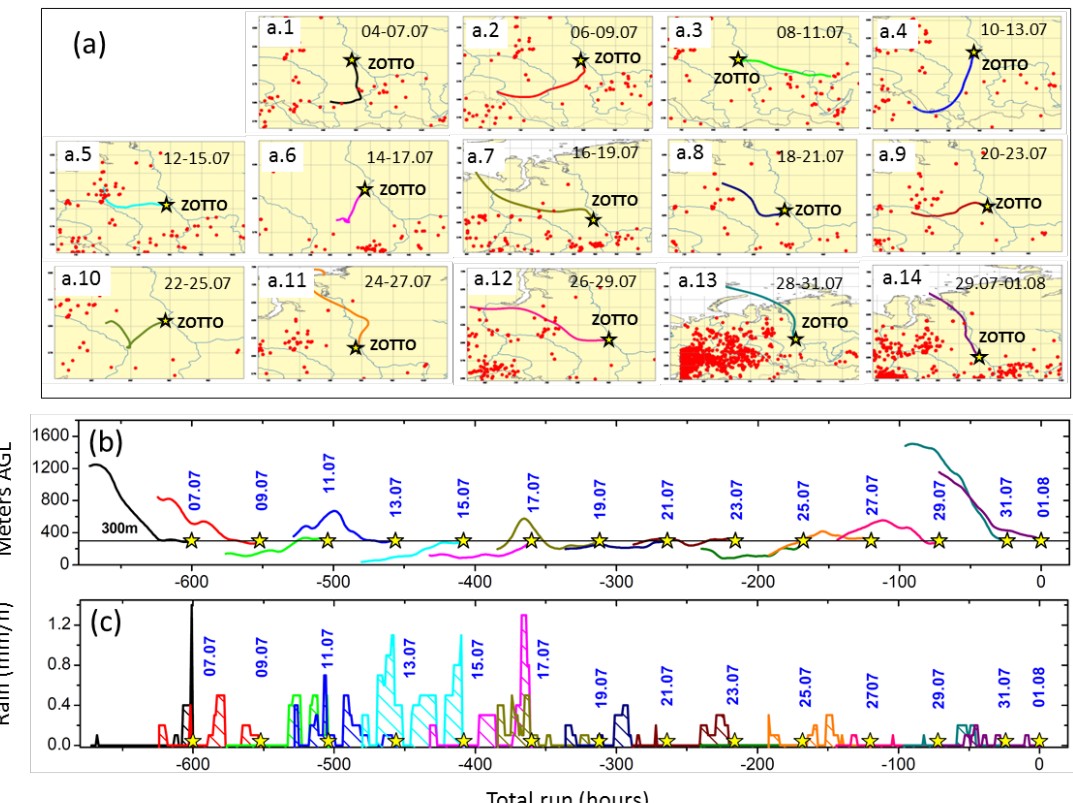

**Figure 12. (a)** 72-h HYSPLIT air mass trajectories with 48-h intervals during background periods at ZOTTO at 300 m from 07 July to 01 August 2010; red dots indicate the fire locations within a 72 h period, **(b)** their height above ground level (AGL), and **(c)** precipitation rate along the air mass trajectories. Stars and dates show the arrival time of the air mass **(b, c)**.

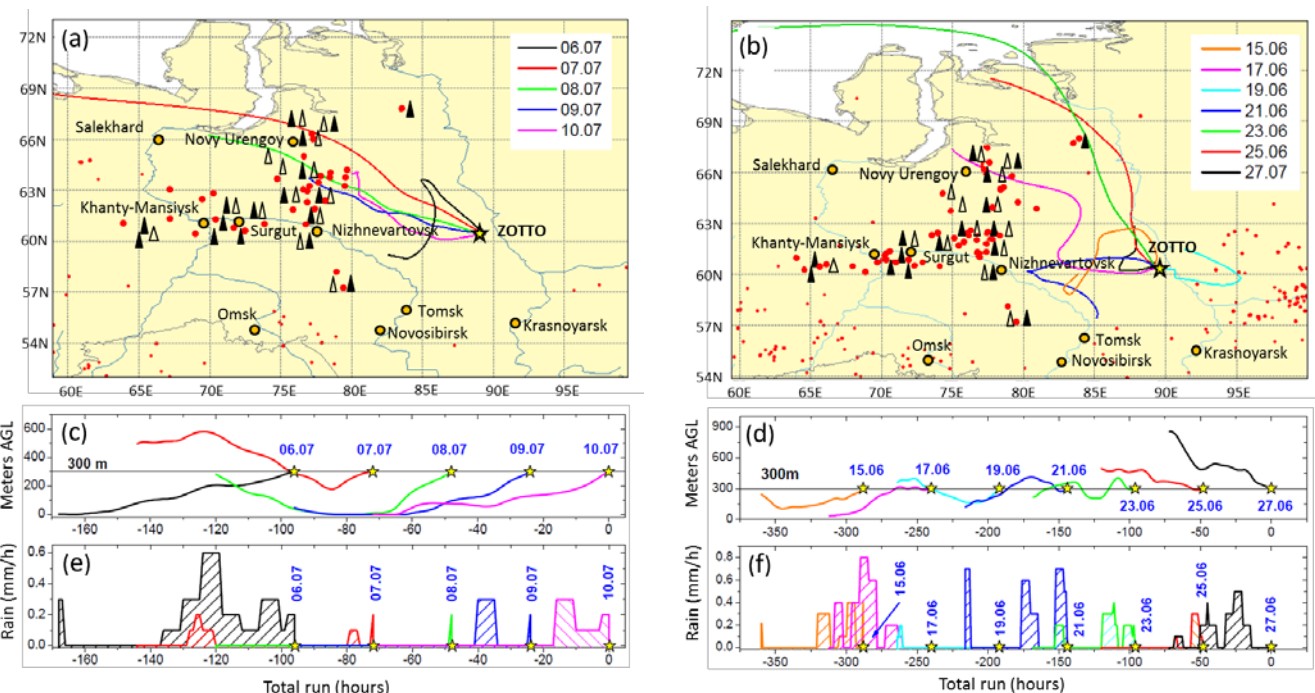

**Figure.13.** 72-h HYSPLIT air mass trajectories during clean periods at ZOTTO at 300 m from 06 to 10 July, 2011 with 24-h intervals (**a**), and from 15 to 27 July 2013 with 48-h intervals (**b**), respectively. Bold red dots indicate the hot spots, which are most likely associated with gas flaring; open and solid triangles stand for gas and oil production fields, respectively; (**b, d**) trajectory heights above ground level (AGL) and (**c, f**) rain rate (mm h$^{-1}$) along the air-mass trajectories. Stars and dates show the arrival times of air masses (**c, d, e,** and **f**).

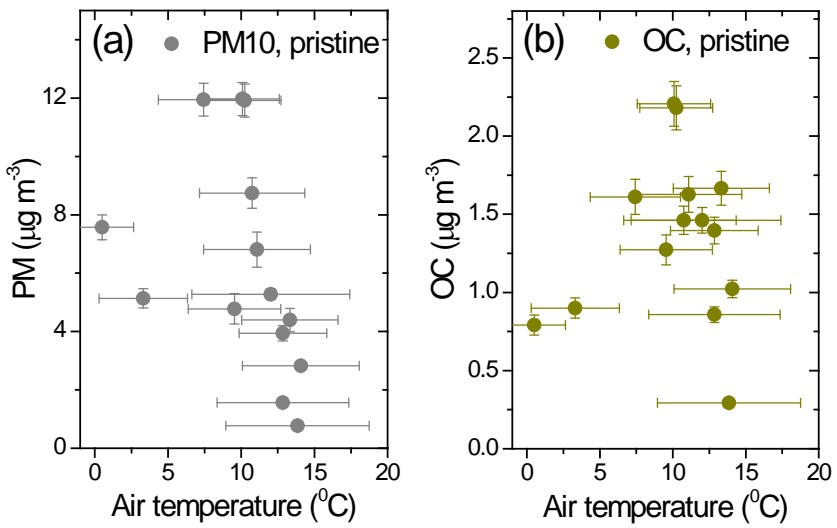

**Figure 14.** PM10 (**a**) and OC (**b**) concentration during pristine summer periods vs. averaged ground-level temperature (± st. dev.) estimated from the meteorological profiles along each trajectory.

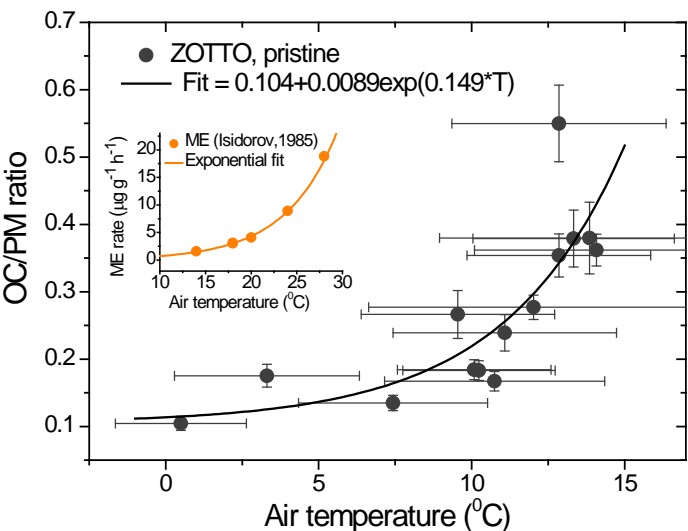

**Figure 15.** OC/PM ratios in PM10 for pristine conditions vs. averaged ground-level temperature (± st. dev.) estimated from the meteorological profiles along each trajectory. The coefficient of determination of the fit ($R^2$) is 0.66. The insert shows monoterpene emission (ME) rate from Scots pine vs. temperature (Isidorov et al., 1985) and the exponential fit: ME rate = $0.121 \cdot \exp[0.179 \cdot T(°C)]$ with $R^2 = 0.99$.