# Peer review of "Long-term measurements (2010 - 2014) of carbonaceous aerosol and carbon monoxide at the Zotino Tall Tower Observatory (ZOTTO) in central Siberia"

_Atmospheric Chemistry and Physics, 2017_

## Referee Comment (RC1) · Anonymous Referee #1 · 1 Jul 2017

This manuscript presents and discusses results from long-term measurements (2010-2014) of the particulate matter (PM) mass, elemental carbon (EC), organic carbon (OC), water-soluble organic carbon (WSOC) for PM10 aerosol filter samples that were collected at the Zotino Tall Tower (ZOTTO) in Central Siberia. The data were complemented with measurements for carbon monoxide (CO). The results extend on previous data (and publications) for the ZOTTO site, such as those given by Chi et al. (2013), which dealt with aerosol and CO data for the period between September 2006 and December 2011. Although the measurements for WSOC could in my opinion have been

done in a better way, I am overall satisfied with the way the measurements were done. I like in particular that a rather novel non-parametric approach named REBS (Robust Extraction of Background Signal) was used to separate pollution and background periods, and that in addition near-pristine air masses were selected based on their EC concentrations being below the detection limit of the thermal/optical instrument used. It definitely made sense to separate the 5-year long time series into polluted, background and clean periods and to analyze and discuss these periods separately. The approach used now seems a clear improvement over that used by Chi et al. (2013).

Nevertheless, as indicated below, the current manuscript suffers from several (mostly technical) shortcomings, so that revision is needed before it can be published in ACP.

Specific comments:

1. Page 4, line 27, and Page 22, line 13: Abbreviations and acronyms should only once be defined (written full-out). "SOA" was already defined on page 2, line 34.

2. Page 8, lines 5-7: It is stated here that "The uncertainty (1 standard deviation) for the PM determination is estimated to be 3.5 $\mu$g for 47-mm quartz filters". Is there any substantiation for this statement? The uncertainty given is unrealistically low. For example, Hitzenberger et al. (Atmos. Environ., 38 (2004) 6467–6476) indicated that "The precision of the net mass determination for 47-mm diameter Whatman QM-A quartz fibre filters is estimated at 30 $\mu$g"; even for 47-mm diameter Nuclepore polycarbonate filters and Gelman Teflo filters (which can be weighed with much better precision), it was stated in the same paper that based on repetitive weightings of blank and loaded filters, the precision (1 standard deviation) of the net mass (i.e., of the particulate mass) was estimated at 5 $\mu$g; incidentally, for 47-mm diameter Millipore mixed cellulose ester filters, the precision was estimated at 30 $\mu$g.

3. Page 15, line 32: Abbreviations and acronyms should only once be defined (written full-out). "APT" was already defined on page 9, line 25.

4. Page 16, lines 25-26: Sodium and potassium cannot be called "compounds"; perhaps, "compounds" can be replaced by "components".

5. Page 18, line 33: It is strange to talk about "receptor sites" for samples that were collected along flights from northern Alaska (Fairbanks).

6. Technical and other (mostly minor) corrections:

- page 1, line 24: replace "10$\mu$m" by "10 $\mu$m".

- page 3, line 29: replace "have light-absorbing" by "have a light-absorbing".

- page 5, line 1: is "Andreae et al., 2008" 2008a or 2008b or perhaps 2008a,b?

- page 5, line 13: replace "Joutsensari, 2015" by "Joutsensari et al., 2015".

- page 7, line 8: replace "10$\mu$m" by "10 $\mu$m".

- page 8, line 30: replace "are most" by "are the most".

- page 8, line 32: replace "NOISH-" by "NIOSH-".

- page 9, line 10: replace "by the two" by "the two".

- page 9, line 22: replace "calculated an" by "calculated for an".

- page 9, lines 30 and 34: replace "Giglio," by "Giglio et al.,".

- page 12, line 12: A Tab should be inserted before "Table 2 shows".

- page 13, line 7: replace "Saarino" by "Saarnio".

- page 15, line 17: A Tab should be inserted before "The lack".

- page 16, line 2: replace "2011(Fig." by "2011 (Fig.".

- page 16, line 15: A Tab should be inserted before "The".

- page 16, lines 17-18, and page 48, Table 3: "Verma et al., 2010" is not in the Reference list.

- page 16, line 22: A Tab should be inserted before "In winter".

- page 17, line 5: replace "in a strong" by "in strong".

- page 17, line 32: A Tab should be inserted before "The large-scale".

- page 18, line 24: A Tab should be inserted before "As a case".

- page 19, line 1: A Tab should be inserted before "In turn".

- page 20, line 7: replace "during selected" by "during the selected".

- page 20, line 23: A Tab should be inserted before "Both CO".

- page 20, line 30: A Tab should be inserted before "In contrast".

- page 20, line 30: replace "Fig.11c" by "Fig. 11c" and replace "Fig.11d" by "Fig. 11d".

- page 21, line 9: "Qi et al., 2016" is not in the Reference list; there is "Qi et al., 2017" in that list to which no reference is made within the text.

- page 22, line 11: replace "et al.,2005" by "et al., 2005".

- page 22, line 12: "Ebben et al., 2014" is not in the Reference list; there is "Ebben et al., 2011" in that list to which no reference is made within the text.

- page 22, line 15: replace "et al.,2014" by "et al., 2014".

- page 22, line 16: replace "Isidorov," by "Isidorov et al.,".

- page 24, line 7: replace "parameter along" by "parameters along".

- page 25, line 11: A Tab should be inserted before "Over the".

- page 25, line 11: replace "remain relatively" by "remains relatively".

- page 25, line 29: replace "during ZOTTO" by "during the ZOTTO".

- pages 26-45, References: For authors with more than 1 initial, there should be a space between the initials; references with the same first author (e.g., Andreae, Birch, Bond, Chi, Heintzenberg, Kanaya, Kulmala, Maenhaut, Mikhailov) should be in the appropriate ACP order; for references with at least 3 authors, there should be ", and" before the last author (note that for references with only 2 authors, there should not be ", and" but " and" instead; the titles of journal articles should be in lower case instead of in Title Case; abbreviated journal names should be used throughout; furthermore, there should be a comma before the abbreviated journal name and there should be a comma followed by a space before the journal volume.

- page 27, line 4: replace "Glob. Biogeochem. Cy." by "Global Biogeochem. Cycles".

- page 28, line 16: replace "Aerosol Sci." by "J. Aerosol Sci.".

- page 29, line 4: replace "Techn." by "Technol.".

- page 29, line 28: replace "Tellus" by "Tellus B".

- page 29, line 32: replace "Paulsen, M., 2006b." by "Paulsen, M.:".

- page 31, line 5: replace "Remote. Sens." by "Remote Sens.".

- page 31, lines 7-8: this reference should come after "Goetz et al. (2007)".

- page 33, lines 7-9: this reference should come after "Liao et al. (2014)".

- page 35, line 14: replace "Biogeochem. Cy." by "Biogeochem. Cycles".

- page 35, lines 27-28: there is not referred to this reference within the text.

- page 35, lines 29-32: this reference should come before "Lack and Cappa (2010)".

- page 37, lines 11-12 and line 15 : replace "Spectrum." by "Spectrom.".

- page 38, line 14: replace "Geoph." by "Geophys.".

- page 38, line 27: replace "Techn." by "Technol.".

- page 38, line 33: replace "Scientific Report" by "Scientific Reports".

- page 40, line 17: replace "Grothe, H., 2012." by "Grothe, H.:".

- page 41, line 22: replace "2016" by "2016.".

- page 41, line 31: replace "Atm." by "Atmos.".

- page 41, lines 33-34: there is not referred to this reference within the text.

- page 42, line 6: replace "Aeros." by "Aerosol".

- page 42, line 9: replace "Pollution" by "Pollut.".

- page 42, line 21: replace "Geoph." by "Geophys.".

- page 42, lines 27-33: these two references should come before "Singh et al. (2016)".

- page 44, lines 14-17: this reference should come before "Vestenius et al. (2014)".

- page 44, line 18 up to page 45, line 4: theses references are not in the proper order; they should be ordered alphabetically according to the first author's first initial.

- page 46, Table 1: replace "Chi et al." by "Chi" (3 times); furthermore, "Qu et al., 2009" and Zhao et al., 2015" are not in the Reference list.

- page 48, Table 3, heading: replace "stands for" by "stand for".

- page 48, Table 3: "Cao et al. (2005)" is not in the Reference list.

- page 49, Table 3. Continued: "McMeeking et al. (2009)" is not in the Reference list.

- page 50, Table 3. Continued: replace "Li et al., 2007" by "Li et al. (2007)".

- page 51, Table 4, heading: replace "The third column is average" by "The third column is the average".

- page 52, Table 5, heading: replace "The average" by "Average".

[Figure]

- page 54, Figure 4, caption: replace "The seasonal" by "Seasonal".

- page 55, Figure 5, caption: replace "The seasonal" by "Seasonal".

- page 57, Figure 11, caption: replace "(b), (c) median" by "(c), (d) median".

---

## Referee Comment (RC2) · Anonymous Referee #2 · 29 Aug 2017

This manuscript presents results from long-term measurements of PM mass, EC, OC, and WSOC concentrations in PM10 filter samples collected at the Zotino Tall Tower (ZOTTO) in Siberia over the period of 5 years (2010-2014). These measurements are also complemented with CO measurements. The manuscript is well written and data has been adequately discussed. However, there are a few shortcomings which needs to be addressed before considering this manuscript for publication.

Specific comments:

(1) Abstract is too long and shall be shortened by 30 or 40%.

(2) Page 8, line 5: how the uncertainty on PM mass determination was assessed? Include this information in the text.

(3) WSOC was measured indirectly, which can be a source of significant uncertainty. The method used here may overestimate WSOC because 12 h soaking of filter in deionized water would remove water-soluble fraction of OC, however, at the same time some insoluble OC may also come out of the filter, which will be estimated as WSOC. How the reliability of this method was tested? This information shall be included in the text.

(4) Uncertainty in WSOC measurements will also affect the WSOC/OC ratio reported in this study, which shall be discussed in the text.

(5) To convert OC into OM, a conversion factor of 1.8 has been used, which is another source of uncertainty in estimated TCA. This conversion factor is likely not uniform throughout the study period. It may vary from 1.4 to 2.2 (Turpin and Lim, AS&T, 2001). This fact shall be mentioned in the text while discussing TCA or TCA/PM data.

(6) Page 10, line 25: WSOC/OC ratios are given in %, which looks odd. Ratio should be written in fraction form e.g. 65% should be written as 0.65.

(7) At many places in the manuscript, sometime abbreviations are used and sometime the full words are used. Use abbreviations only after defining them when they appear for the first time.

(8) Fig. 3: y-axis is on log scale. Start y-axis from 1 rather than 0.1 so that EnF are prominently visible.

(9) Interpretation of Fig. 15 doesn't look very convincing as the there is no significant effect of temperature on monoterpene emissions upto 15 oC or so, whereas OC/PM ratio shows the observed trend for the temperature ranging from 0 to 15 oC.

---

## Author Comment (AC1) · 9 Oct 2017

Response to Anonymous Referee #1

The referee's comments are in italics, our responses in plain font.

*This manuscript presents and discusses results from long-term measurements (2010-2014) of the particulate matter (PM) mass, elemental carbon (EC), organic carbon (OC), water-soluble organic carbon (WSOC) for PM10 aerosol filter samples that were collected at the Zotino Tall Tower (ZOTTO) in Central Siberia. The data were complemented with measurements for carbon monoxide (CO). The results extend on previous data (and publications) for the ZOTTO site, such as those given by Chi et al. (2013), which dealt with aerosol and CO data for the period between September 2006 and December 2011. Although the measurements for WSOC could in my opinion have been done in a better way, I am overall satisfied with the way the measurements were done. I like in particular that a rather novel non-parametric approach named REBS (Robust Extraction of Background Signal) was used to separate pollution and background periods, and that in addition near-pristine air masses were selected based on their EC concentrations being below the detection limit of the thermal/optical instrument used. It definitely made sense to separate the 5-year long time series into polluted, background and clean periods and to analyze and discuss these periods separately. The approach used now seems a clear improvement over that used by Chi et al. (2013). Nevertheless, as indicated below, the current manuscript suffers from several (mostly technical) shortcomings, so that revision is needed before it can be published in ACP.*

We thank the Referee #1 for the constructive criticism and suggestions for improvement that were taken into account upon manuscript revision. Responses to individual comments are given below.

*Specific comments:*
*1. Page 4, line 27, and Page 22, line 13: Abbreviations and acronyms should only once be defined (written full-out). "SOA" was already defined on page 2, line 34.*

Done

*2. Page 8, lines 5-7: It is stated here that "The uncertainty (1 standard deviation) for the PM determination is estimated to be 3.5 μg for 47-mm quartz filters". Is there any substantiation for this statement? The uncertainty given is unrealistically low. For example, Hitzenberger et al. (Atmos. Environ., 38 (2004) 6467–6476) indicated that "The precision of the net mass determination for 47-mm diameter Whatman QM-A quartz fibre filters is estimated at 30 μg"; even for 47-mm diameter Nuclepore polycarbonate filters and Gelman Teflo filters (which can be weighed with much better precision), it was stated in the same paper that based on repetitive weightings of blank and loaded filters, the precision (1 standard deviation) of the net mass (i.e., of the particulate mass) was estimated at 5 μg; incidentally, for 47-mm diameter Millipore mixed cellulose ester filters, the precision was estimated at 30 μg.*

We re-checked the uncertainty of the PM determination using a Mettler-Toledo micro balance model XP6. The Table below shows typical weighing results from four aerosol-loaded quartz filters. The same measurement uncertainty (even better) was observed for the back filters. Based on the gravimetric measurements the PM uncertainty was estimated at 10 μg. Thus, we replaced the initial value of 3.5 μg by 10 μg.

Table. Weighing protocol (mg)

| | | | | |
|---|---|---|---|---|
| 112.386 | 110.787 | 111.968 | 116.159 | |
| 112.381 | 110.786 | 111.948 | 116.167 | |
| 112.397 | 110.798 | 111.964 | 116.174 | |
| **112.388** | **110.790** | **111.960** | **116.167** | **Aver** |
| **0.0067** | **0.0054** | **0.0086** | **0.0061** | **Stdev** |

*3. Page 15, line 32: Abbreviations and acronyms should only once be defined (written full-out). "APT" was already defined on page 9, line 25.*

Done

*4. Page 16, lines 25-26: Sodium and potassium cannot be called "compounds"; perhaps, "compounds" can be replaced by "components".*

Done

*5. Page 18, line 33: It is strange to talk about "receptor sites" for samples that were collected along flights from northern Alaska (Fairbanks).*

The "receptor sites" are replaced by "sampling areas".

*6. Technical and other (mostly minor) corrections:*
*- page 1, line 24: replace "10$\mu m$" by "10 $\mu m$".*
*- page 3, line 29: replace "have light-absorbing" by "have a light-absorbing".*
*- page 5, line 1: is "Andreae et al., 2008" 2008a or 2008b or perhaps 2008a,b?*
*- page 5, line 13: replace "Joutsensari, 2015" by "Joutsensari et al., 2015".*
*- page 7, line 8: replace "10$\mu m$" by "10 $\mu m$".*
*- page 8, line 30: replace "are most" by "are the most".*
*- page 8, line 32: replace "NOISH-" by "NIOSH-".*
*- page 9, line 10: replace "by the two" by "the two".*
*- page 9, line 22: replace "calculated an" by "calculated for an".*
*- page 9, lines 30 and 34: replace "Giglio," by "Giglio et al.,".*
*- page 12, line 12: A Tab should be inserted before "Table 2 shows".*
*- page 13, line 7: replace "Saarino" by "Saarnio".*
*- page 15, line 17: A Tab should be inserted before "The lack".*
*- page 16, line 2: replace "2011(Fig." by "2011 (Fig.".*
*- page 16, line 15: A Tab should be inserted before "The".*
*- page 16, lines 17-18, and page 48, Table 3: "Verma et al., 2010" is not in the Reference list.*
The reference is added.

*- page 16, line 22: A Tab should be inserted before "In winter".*
*- page 17, line 6: replace "in a strong" by "in strong".*
*- page 17, line 32: A Tab should be inserted before "The large-scale".*
*- page 18, line 24: A Tab should be inserted before "As a case".*
*- page 19, line 1: A Tab should be inserted before "In turn".*
*- page 20, line 7: replace "during selected" by "during the selected".*
*- page 20, line 23: A Tab should be inserted before "Both CO".*
*- page 20, line 30: A Tab should be inserted before "In contrast".*
*- page 20, line 30: replace "Fig.11c" by "Fig. 11c" and replace "Fig.11d" by "Fig. 11d".*
*- page 21, line 9: "Qi et al., 2016" is not in the Reference list; there is "Qi et al., 2017" in that list to which no reference is made within the text.*
The link is corrected.

*- page 22, line 11: replace "et al.,2005" by "et al., 2005".*
*- page 22, line 12: "Ebben et al., 2014" is not in the Reference list; there is "Ebben et al., 2011" in that list to which no reference is made within the text.*
The link is corrected.

*- page 22, line 15: replace "et al.,2014" by "et al., 2014".*
*- page 22, line 16: replace "Isidorov," by "Isidorov et al.,".*
*- page 24, line 7: replace "parameter along" by "parameters along".*
*- page 25, line 11: A Tab should be inserted before "Over the".*
*- page 25, line 11: replace "remain relatively" by "remains relatively".*
*- page 25, line 29: replace "during ZOTTO" by "during the ZOTTO".*
*- pages 26-45, References: For authors with more than 1 initial, there should be a space between the initials; references with the same first author (e.g., Andreae, Birch,*

*Bond, Chi, Heintzenberg, Kanaya, Kulmala, Maenhaut, Mikhailov) should be in the appropriate ACP order; for references with at least 3 authors, there should be ", and" before the last author (note that for references with only 2 authors, there should not be ", and" but " and" instead; the titles of journal articles should be in lower case instead of in Title Case; abbreviated journal names should be used throughout; furthermore, there should be a comma before the abbreviated journal name and there should be a comma followed by a space before the journal volume.*

*- page 27, line 4: replace "Glob. Biogeochem. Cy." by "Global Biogeochem. Cycles".*
*- page 28, line 16: replace "Aerosol Sci." by "J. Aerosol Sci.".*
*- page 29, line 4: replace "Techn." by "Technol.".*
*- page 29, line 28: replace "Tellus" by "Tellus B".*
*- page 29, line 32: replace "Paulsen, M., 2006b." by "Paulsen, M.:".*
*- page 31, line 5: replace "Remote. Sens." by "Remote Sens.".*
*- page 31, lines 7-8: this reference should come after "Goetz et al. (2007)".*
*- page 33, lines 7-9: this reference should come after "Liao et al. (2014)".*
*- page 35, line 14: replace "Biogeochem. Cy." by "Biogeochem. Cycles".*
*- page 35, lines 27-28: there is not referred to this reference within the text.*
*- page 35, lines 29-32: this reference should come before "Lack and Cappa (2010)".*
*- page 37, lines 11-12 and line 15 : replace "Spectrum." by "Spectrom.".*
*- page 38, line 14: replace "Geoph." by "Geophys.".*
*- page 38, line 27: replace "Techn." by "Technol."*

*- page 38, line 33: replace "Scientific Report" by "Scientific Reports".*
*- page 40, line 17: replace "Grothe, H., 2012." by "Grothe, H.:".*
*- page 41, line 22: replace "2016" by "2016.".*
*- page 41, line 31: replace "Atm." by "Atmos.".*
*- page 41, lines 33-34: there is not referred to this reference within the text.*
The reference has been removed.

*- page 42, line 6: replace "Aeros." by "Aerosol".*
*- page 42, line 9: replace "Pollution" by "Pollut.".*
*- page 42, line 21: replace "Geoph." by "Geophys.".*
*- page 42, lines 27-33: these two references should come before "Singh et al. (2016)".*
*- page 44, lines 14-17: this reference should come before "Vestenius et al. (2014)".*
*- page 44, line 18 up to page 45, line 4: theses references are not in the proper order; they should be ordered alphabetically according to the first author's first initial.*
*- page 46, Table 1: replace "Chi et al." by "Chi" (3 times); furthermore, "Qu et al., 2009" and Zhao et al., 2015" are not in the Reference list.*
The references to Qu et al.,2009 and Zhao et al., 2015 are added.

*- page 48, Table 3, heading: replace "stands for" by "stand for".*
*- page 48, Table 3: "Cao et al. (2005)" is not in the Reference list.*
*- page 49, Table 3. Continued: "McMeeking et al. (2009)" is not in the Reference list.*
The references to Cao et al., 2005 and McMeeking et al., 2009 are added.

*- page 50, Table 3. Continued: replace "Li et al., 2007" by "Li et al. (2007)".*
*- page 51, Table 4, heading: replace "The third column is average" by "The third column is the average".*
*- page 52, Table 5, heading: replace "The average" by "Average".*
*- page 54, Figure 4, caption: replace "The seasonal" by "Seasonal".*
*- page 55, Figure 5, caption: replace "The seasonal" by "Seasonal".*
*- page 57, Figure 11, caption: replace "(b), (c) median" by "(c), (d) median".*

All the above reviewer's comments are taken into account.

---

## Author Comment (AC2)

Response to Anonymous Referee #2

The referee's comments are in italics, our responses in plain font.

*This manuscript presents results from long-term measurements of PM mass, EC, OC, and WSOC concentrations in PM10 filter samples collected at the Zotino Tall Tower (ZOTTO) in Siberia over the period of 5 years (2010-2014). These measurements are also complemented with CO measurements. The manuscript is well written and data has been adequately discussed. However, there are a few shortcomings which needs to be addressed before considering this manuscript for publication.*

We thank the Referee #2 for the constructive criticism and suggestions for improvement that were taken into account upon manuscript revision. Responses to individual comments are given below.

*Specific comments:*
*(1) Abstract is too long and shall be shortened by 30 or 40%.*

Done

*(2) Page 8, line 5: how the uncertainty on PM mass determination was assessed? Include this information in the text.*

We re-checked the uncertainty of the PM determination using a Mettler-Toledo micro balance model XP6. The Table below shows typical weighing results from four aerosol-loaded quartz filters. The same measurement uncertainty (even better) was observed for the back filters. Based on the gravimetric measurements the PM uncertainty was estimated at 10 µg. Thus, we replaced the initial value of 3.5 µg by 10 µg.

Table.  Weighing protocol (mg)

| | | | | |
|---|---|---|---|---|
| 112.386 | 110.787 | 111.968 | 116.159 | |
| 112.381 | 110.786 | 111.948 | 116.167 | |
| 112.397 | 110.798 | 111.964 | 116.174 | |
| **112.388** | **110.790** | **111.960** | **116.167** | **Aver** |
| **0.0067** | **0.0054** | **0.0086** | **0.0061** | **Stdev** |

*(3) WSOC was measured indirectly, which can be a source of significant uncertainty. The method used here may overestimate WSOC because 12 h soaking of filter in deionized water would remove water-soluble fraction of OC, however, at the same time some insoluble OC may also come out of the filter, which will be estimated as WSOC. How the reliability of this method was tested? This information shall be included in the text.*

The reliability of the method for WSOC analysis was tested by comparison with a TOC analyzer (TOC-$V_{CPH}$. Shimadzu), which allow to determine WSOC in the aqueous extracts directly.  For this method aqueous extracts were prepared by manual shaking of the filter punches during 5 min, after which it was allowed to stand for 30 min. Figure 1 shows that both methods are in agreement, in spite of the fact that extraction time was different (12 h versus 30 min). A good

consistency between TOT and TOC-V$_{CPH}$ methods were also obtained by Timonen et al. (Boreal Environ. Res. 13, 335-346, 2008; Fig. 2).

[Figure]

Figure 1. Comparison of measured WSOC concentrations between TOT and TOC-V$_{CPH}$ method.

The following clarifying text has been added:

In addition to the TOT method, a TOC-V$_{CPH}$ analyzer (5000 A, Shimadzu) was also used for WSOC analysis. A two-step procedure consisting of measurements of water-soluble total carbon (WSTC) and water soluble inorganic carbon (WSIC) was applied. WSOC is then calculated as a difference between WSTC and WSIC (Chi et al., 2009). The TOT and TOC-V$_{CPH}$ measurements of WSOC concentrations cover the date range from April 2010 to December 2011. In general, the agreement between the two methods during this time period was within 10%. Due to fatal technical problems with the TOC-V$_{CPH}$, after December 2011 WSOC was measured only by the Sunset instrument. The estimated error of the WSOC concentrations using the TOT method is 10% - 15%, depending on the filter loading, which results in a 12-17% error for the WSOC/OC ratio.

*(4) Uncertainty in WSOC measurements will also affect the WSOC/OC ratio reported in this study, which shall be discussed in the text.*

This information has been added to the text (see previous response).

*(5) To convert OC into OM, a conversion factor of 1.8 has been used, which is another source of uncertainty in estimated TCA. This conversion factor is likely not uniform throughout the study period. It may vary from 1.4 to 2.2 (Turpin and Lim, AS&T, 2001). This fact shall be mentioned in the text while discussing TCA or TCA/PM data.*

The following text has been added:

Organic matter (OM) was estimated as 1.8·OC. The same OC-to-OM conversion factor of 1.8 had been used in the SMEARII (Finland) (Maenhaut et al., 2011a) and K-puszta (Hungary) (Maenhaut et al., 2008) remote coniferous forest sites, providing the best agreement in the aerosol chemical mass closure calculations. As a result, the total carbonaceous matter (TCM) was calculated as TCM = 1.8·OC+EC. It should be noted that there is considerable variability in reported OM/OC ratios for organic compounds depending on the relative contribution of primary and secondary organic aerosol sources, with reported values ranging from 1.2–2.4 (Turpin and Lim, 2001). In this study OM and TCM are estimated and used mainly to illustrate their temporal variability. However, as will be shown below, the obtained estimates of the TCM/PM10 ratio are reasonably consistent with published values for the sources of the pollution plumes.

*(6) Page 10, line 25: WSOC/OC ratios are given in %, which looks odd. Ratio should be written in fraction form e.g. 65% should be written as 0.65.*

Done

*(7) At many places in the manuscript, sometime abbreviations are used and sometime the full words are used. Use abbreviations only after defining them when they appear for the first time.*

Done

*(8) Fig. 3: y-axis is on log scale. Start y-axis from 1 rather than 0.1 so that EnF are prominently visible.*

Done

*(9) Interpretation of Fig. 15 doesn't look very convincing as the there is no significant effect of temperature on monoterpene emissions up to 15 $^{o}$C or so, whereas OC/PM ratio shows the observed trend for the temperature ranging from 0 to 15 $^{o}$C.*

We do not agree. Our interpretation of the observed near exponential growth of OC/PM ratio due to biogenic activity seems quite reasonable.